# A large-scale genome-wide cross-trait analysis reveals shared genetic architecture between Alzheimer's disease and gastrointestinal tract disorders

Emmanuel O. Adewuyi [1,2✉], Eleanor K. O'Brien [1,2], Dale R. Nyholt [3], Tenielle Porter[1,2,4] & Simon M. Laws [1,2,4✉]

Consistent with the concept of the gut-brain phenomenon, observational studies suggest a relationship between Alzheimer's disease (AD) and gastrointestinal tract (GIT) disorders; however, their underlying mechanisms remain unclear. Here, we analyse several genome-wide association studies (GWAS) summary statistics (N = 34,652–456,327), to assess the relationship of AD with GIT disorders. Findings reveal a positive significant genetic overlap and correlation between AD and gastroesophageal reflux disease (GERD), peptic ulcer disease (PUD), gastritis-duodenitis, irritable bowel syndrome and diverticulosis, but not inflammatory bowel disease. Cross-trait meta-analysis identifies several loci ($P_{meta-analysis} < 5 \times 10^{-8}$) shared by AD and GIT disorders (GERD and PUD) including *PDE4B*, *BRINP3*, *ATG16L1*, *SEMA3F*, *HLA-DRA*, *SCARA3*, *MTSS2*, *PHB*, and *TOMM40*. Colocalization and gene-based analyses reinforce these loci. Pathway-based analyses demonstrate significant enrichment of lipid metabolism, autoimmunity, lipase inhibitors, PD-1 signalling, and statin mechanisms, among others, for AD and GIT traits. Our findings provide genetic insights into the gut-brain relationship, implicating shared but non-causal genetic susceptibility of GIT disorders with AD's risk. Genes and biological pathways identified are potential targets for further investigation in AD, GIT disorders, and their comorbidity.

[1] Centre for Precision Health, Edith Cowan University, Joondalup, Perth, WA 6027, Australia. [2] Collaborative Genomics and Translation Group, School of Medical and Health Sciences, Edith Cowan University, Joondalup, Perth, WA 6027, Australia. [3] Centre for Genomics and Personalised Health, School of Biomedical Sciences, Faculty of Health, Queensland University of Technology, Brisbane, QLD, Australia. [4] Curtin Health Innovation Research Institute, Curtin University, Bentley, Perth, WA 6102, Australia. ✉email: e.adewuyi@ecu.edu.au; s.laws@ecu.edu.au

Alzheimer's disease (AD) is the most prevalent form of dementia, characterised by neurodegeneration and a progressive decline in cognitive ability[1,2]. The disorder ranks as a subject of increasing global public health importance with consequences for wide-ranging social and economic adverse impacts on sufferers, their families, and the society at large[1]. By the year 2030, over 82 million people—and about 152 million by 2050—are projected to suffer from AD[1,2]. While AD has no known curative treatments, and its pathogenesis is yet to be clearly understood, a comprehensive assessment of its shared genetics with other diseases (comorbidities) can provide a deeper understanding of its underlying biological mechanisms and enhance potential therapy development efforts.

Several studies have reported a pattern of co-occurrence of dementia (and AD in particular) with certain gastrointestinal tract (GIT) disorders, microbiota, dysbiosis or medications commonly used in the treatment of peptic ulcer disease (PUD)[3–10]. For example, an observational study reported more than twice the odds of dementia in individuals with gastritis (adjusted odds ratio [AOR]: 2.42, $P < 0.001$, 95% confidence interval [CI]: 1.68–3.49)[3]. Another observational study found a significant association between regular use of proton-pump inhibitors (PPI, medications for gastritis duodenitis, gastro-esophageal reflux disease [GERD] or PUD) and increased risk of incident dementia (hazard ratio [HR]: 1.44 [95% CI, 1.36–1.52]; $P < 0.001$)[4]. Similarly, lansoprazole (a PPI) was reported to promote amyloid-beta (Aβ) production[5], the accumulation of which is central to one of the core hypotheses for the development of AD[11]. More recently, a longitudinal study reported more than a sixfold increased risk of AD in individuals with inflammatory bowel disease (IBD) [HR: 6.19, 95%CI: 3.31–11.57], predicting over five-fold increased incidence across all forms of dementia[7].

The available evidence, thus, suggests comorbidity or some forms of association between AD and GIT disorders, although it is not clear whether GIT traits are risks for AD or vice versa. Regardless, these findings agree with the concept of the 'gut–brain' axis or the 'gastric mucosa–brain' relationship, which has been implicated between GIT-related traits and central nervous system (CNS) disorders including depression and Parkinson's disease[12–17]. A relationship between AD and GIT disorders or their comorbidity can worsen the quality of life of sufferers while contributing to increased healthcare costs.

Despite the increasing number of studies reporting an association between AD and GIT traits, the biological mechanism(s) underlying this potential association remains unclear. Moreover, contrasting evidence exists[7,18,19], leading to a longstanding debate on the potential links of GIT traits to the risk of AD[15,18–20]. Large-scale genome-wide association studies (GWAS), identifying an increasing number of single nucleotide polymorphism (SNPs), genes, and susceptibility loci, have been conducted separately for AD and a range of GIT traits[21–24]. Findings from these GWAS provide compelling evidence for the roles of genetics in the aetiologies of AD and GIT disorders including GERD, PUD, PGM (a combination of disease-diagnosis of PUD and/or GERD and/or corresponding medications and treatments—a potential proxy for PUD or GERD), gastritis-duodenitis, irritable bowel syndrome (IBS), diverticular disease, and IBD[21–24]. However, to the best of our knowledge, no study has leveraged the possible pleiotropy between AD and GIT disorders as a basis for discovering their shared SNPs, genes and/or susceptibility loci.

In this study, we analyse well-powered GWAS summary data to comprehensively assess the genetic relationship and potential causal association between AD and GIT disorders. We demonstrate a positive significant genetic overlap and correlation between AD and GERD, PUD, PGM, IBS, gastritis-duodenitis, and diverticular disease. Also, in a cross-trait GWAS meta-

analysis, we identify many loci shared by AD and GIT disorders. Causality assessment reveals no evidence for a significant causal association between AD and GIT disorders. However, we identify shared genes reaching genome-wide significance for AD and GIT disorders in gene-based association analyses. Lastly, pathway-based analyses show significant enrichment of lipid metabolism, autoimmunity, lipase inhibitors, PD-1 signalling and statin mechanisms, among others, for AD and GIT traits.

## Results

Figure 1 presents a schematic workflow for this study. Briefly, we performed three broad levels of analyses—SNP-level, gene-level, and pathway-based analyses. First, we used the linkage disequilibrium score regression (LDSC)[25] to estimate the genetic correlation between AD and GIT traits, and the 'SNP effect concordance analysis' (SECA)[26] method for concordance in SNP risk effect assessment. Second, to identify SNPs and susceptibility loci shared by AD and GIT disorders, we carried out GWAS meta-analyses. We also applied the pairwise GWAS (colocalisation) method[27] to identify independent genomic loci with shared genetic influence on AD and GIT disorders. Third, using the Mendelian randomisation (MR)[28] and the Latent Causal Variable (LCV)[29] methods, we assessed potential (and partial) causal associations between AD and GIT disorders. Lastly, we performed gene and pathway-based analyses to identify shared genes reaching genome-wide significance and biological pathways for AD and GIT disorders. The largest publicly available AD summary statistics and GIT summary data from research consortia or public repositories were utilised for analysis (Table 1 and Supplementary Data 1).

**Genetic correlation between AD and GIT disorders**. We assessed and quantified the SNP-level genetic correlation between AD and GIT disorders using the LDSC[25] analysis method. The apolipoprotein E (APOE) region has a large effect on the risk of AD; hence, we excluded APOE and the 500 kilobase (kb) flanking region (hg19, 19:44,909,039–45,912,650) from the AD GWAS. We also excluded SNPs in the 26 to 36 megabase region of chromosome six from the data given the complex LD structure in the human major histocompatibility complex (MHC). Notably, in analyses both with and without the APOE region, LDSC reveals a significant genetic correlation between AD and GIT traits (Table 2). Genetic covariance intercept estimates were not significantly different from zero (Supplementary Data 2), indicating no sample overlap between our AD and GIT GWAS.

We found a positive and significant genetic correlation ($r_g$) of AD (excluding APOE region) with GERD ($r_g = 0.25$, $P = 8.19 \times 10^{-18}$), PUD ($r_g = 0.28$, $P = 3.70 \times 10^{-7}$), PGM ($r_g = 0.22$, $P = 2.38 \times 10^{-14}$), gastritis-duodenitis ($r_g = 0.24$, $P = 2.40 \times 10^{-8}$), IBS ($r_g = 0.19$, $P = 1.10 \times 10^{-4}$), and diverticular disease ($r_g = 0.15$, $P = 2.97 \times 10^{-5}$). However, we found no evidence of a significant genetic correlation between AD and IBD ($r_g = 0.07$, $P = 9.94 \times 10^{-2}$) [Table 2], which may be because of the relatively small cases and sample size of the IBD GWAS (Table 1 and Supplementary Data 1). Our estimates of effective sample size (Supplementary Data 1) suggest the IBD GWAS was underpowered compared to other GIT data sets. We reproduced a pattern of a positive and significant genetic correlation between AD[21] and the replication set of GIT traits with or without the APOE region, except for IBD (Supplementary Data 3).

**SNP effect concordance analysis (SECA) results**. Using the SECA method[26], we assessed the directions of SNP-level genetic overlap between AD and GIT disorders. We provide a more comprehensive description of SECA in the methods section.

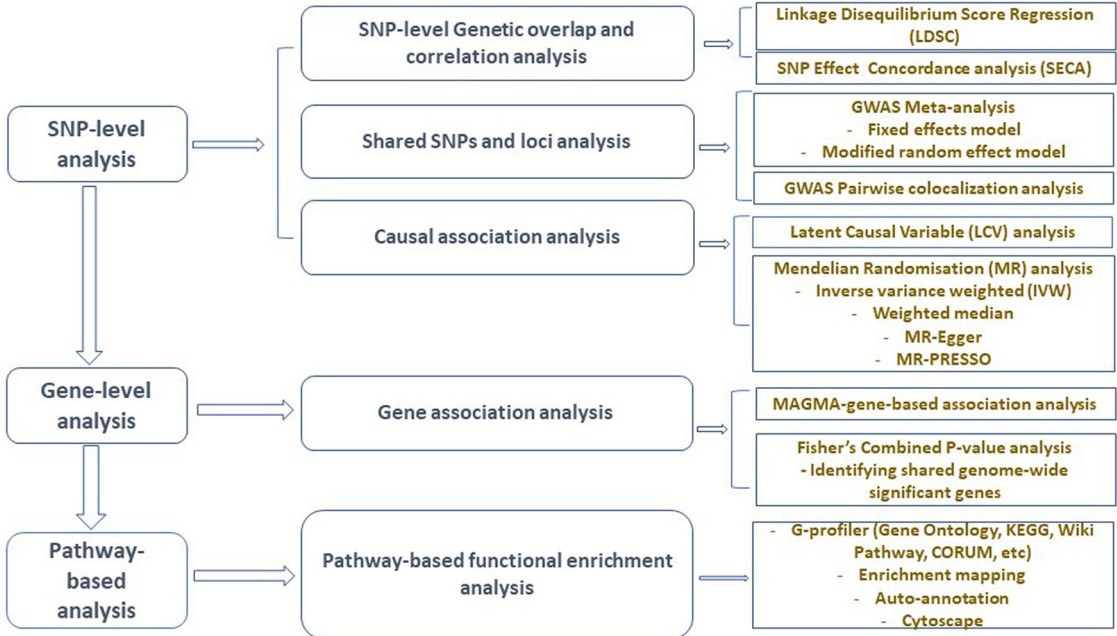

**Fig. 1 Study design and workflow: examining shared genetic and causality of GIT disorders with the risk of Alzheimer's disease.** GWAS genome-wide association studies, SNP single-nucleotide polymorphism, SECA SNP effect concordance analysis, LDSC linkage disequilibrium score regression, LCV latent causal variable, MAGMA multi-marker analysis of genomic annotation, MR Mendelian randomisation, MR-PRESSO Mendelian randomisation pleiotropy residual sum and outlier, KEGG Kyoto Encyclopedia of Genes and Genomes.

**Table 1 Summary of GWAS data sets analysed.**

| GWAS summary statistics | Cases | Control | Sample size | Ancestry | Phenotype source/definition |
|---|---|---|---|---|---|
| AD (Jansen et al.[21]) | 71,880 | 383,378 | 455,258 | European | Clinically diagnosed and UKB AD-by-proxy[21] |
| GERD-UKBB_QSKIN (An et al.[23]) | 71,522 | 261,079 | 332,601 | | Data from the UKB and the QSKIN study[23] |
| PUD (Wu et al.[22]) | 16,666 | 439,661 | 456,327 | | UKB data code described in Wu et al.[22,a] |
| PGM (Wu et al.[22]) | 90,175 | 366,152 | 456,327 | | GWAS for diagnosis of PUD and/or GERD and/or corresponding medications and treatments from the UKB data[22,a] |
| Gastritis-duodenitis Phecode 535 (Lee Lab) | 28,941 | 378,124 | 407,065 | | Full European data subset from the Lee Lab |
| IBS (Wu et al.[22]) | 28,518 | 426,803 | 455,321 | | UKB data code described in Wu et al.[22,a] |
| Diverticulosis Phecode 562 (Lee Lab) | 27,311 | 334,783 | 362,094 | | Full European data subset from the Lee Lab |
| IBD (Wu et al.[22]) | 7045 | 449,282 | 456,327 | | UKB data code described in Wu et al.[22,a] |
| Replication set | | | | | |
| GORD (Wu et al.[22]) | 54,854 | 401,473 | 456,327 | European | UKB data code described in Wu et al.[22,a] |
| PUD Phecode 531 (Lee Lab) | 7436 | 401,525 | 408,961 | | Full European data subset from the Lee Lab |
| Lansoprazole (Watanabe et al.[91]) | 13,559 | 266,884 | 280,443 | | UKB treatment/medication code: lansoprazole[91] |
| Gastritis-duodenitis (Watanabe et al.[91]) | 14,477 | 286,314 | 300,791 | | Main ICD10: K29 Gastritis and duodenitis[91] |
| IBS Phecode 564.1 (Lee Lab) | 5548 | 334,783 | 340,331 | | Full European data subset from the Lee Lab |
| Diverticular disease (Watanabe et al.[91]) | 14,028 | 286,763 | 300,791 | | Main ICD10: K57 Diverticular disease[91] |
| IBD (Liu et al.[49]) | 12,882 | 21,770 | 34,652 | | Data from the IBD genetic consortium[49] |

The 'clinically diagnosed AD' combined data from three case–control cohorts ($N = 79,145$). 'AD-by proxy' data were based on the UKB phenotype definition of individuals whose biological parents were affected by AD. The parent's current age, and where relevant, age at death were reported along with this GWAS data. The genetic correlation between the 'clinically diagnosed AD' and the 'AD-by proxy' is high at 0.81[21], providing strong evidence or justification for combining them as more comprehensively described in the associated publication[21].
AD Alzheimer's disease, GERD and GORD gastroesophageal reflux disease, PUD peptic ulcer disease, PGM GWAS combining disease-diagnosis of PUD and/or GERD and/or medications for their treatments, IBS irritable bowel disease, IBD inflammatory bowel disease, ICD International Classification of Diseases, UKB United Kingdom Biobank.
[a]UKB data code for case definition was from death register, primary care, hospital admissions data, self-report only, and other sources as described in the original publication Wu et al.[22]. The replication set data were used for reproducibility testing in LDSC and SECA analyses, and partly in LCV analysis.

Briefly, SECA performs a bi-directional analysis, assessing concordance in the direction of the effect of AD-associated SNPs (data set 1) on each of the GIT disorders (data set 2) and vice versa. First, we conducted two rounds of P-value informed LD clumping (first clumping: -clump-r² 0.1, -clump-kb 1000; second clumping: -clump-r² 0.1, -clump-kb 10000) using PLINK 1.90[30].

SECA subsequently assesses (using Fisher's test) the presence of excess SNPs in which the direction of effects is concordant across 144 subsets of data set 1 (AD GWAS) and data set 2 (each of the GIT traits GWAS).

We found a positive and significant concordance of SNP risk effect across the AD (data set 1) and each of the GIT GWAS (data

**Table 2 Genetic correlation between AD and GIT disorders.**

| Trait 1 | Trait 2 | $r_g$ | se | P |
|---|---|---|---|---|
| AD (Jansen et al.[21]) | GERD-UKBB_QSKIN (An et al.)[23] | 0.25 | 0.03 | $8.19 \times 10^{-18}$ |
| Excluding *APOE* and MHC regions | PUD (Wu et al.[22]) | 0.28 | 0.05 | $3.70 \times 10^{-7}$ |
| | PGM (Wu et al.[22]) | 0.22 | 0.03 | $2.38 \times 10^{-14}$ |
| | Gastritis-duodenitis Phecode 535 | 0.24 | 0.04 | $2.40 \times 10^{-8}$ |
| | IBS (Wu et al.[22]) | 0.19 | 0.05 | $1.10 \times 10^{-4}$ |
| | Diverticulosis Phecode 562 | 0.15 | 0.04 | $2.97 \times 10^{-5}$ |
| | IBD (Wu et al.[22]) | 0.07 | 0.05 | $9.94 \times 10^{-2}$ |
| AD (Jansen et al.[21]) with *APOE* region | GERD-UKBB_QSKIN An et al 2019[23] | 0.23 | 0.04 | $1.20 \times 10^{-10}$ |
| | PUD (Wu et al.[22]) | 0.26 | 0.05 | $4.25 \times 10^{-7}$ |
| | PGM (Wu et al.[22]) | 0.21 | 0.04 | $3.56 \times 10^{-9}$ |
| | Gastritis-duodenitis Phecode 535 | 0.22 | 0.05 | $1.21 \times 10^{-5}$ |
| | IBS (Wu et al.[22]) | 0.18 | 0.05 | $7.61 \times 10^{-5}$ |
| | Diverticulosis Phecode 562 | 0.14 | 0.03 | $6.58 \times 10^{-5}$ |
| | IBD (Wu et al.[22]) | 0.06 | 0.04 | $1.72 \times 10^{-1}$ |

We applied Bonferroni adjustment for testing the effects of seven GIT traits on AD ($0.05/7 = 7.1 \times 10^{-3}$), and all genetic correlation results surviving this cut-off were considered significant while those having $P < 0.05$ were regarded nominally significant.
*AD* Alzheimer's disease, *GIT* gastrointestinal tract, *GERD* gastroesophageal reflux disease, *PUD* peptic ulcer disease, *IBS* irritable bowel syndrome, *PGM* GWAS combining disease-diagnosis of PUD and/or GERD and/or medications for their treatments, *IBD* inflammatory bowel disease, $r_g$ genetic correlation, *se* standard error, *P* P value, *MHC* major histocompatibility complex.

**Table 3 SECA results: primary test for concordant SNP effects.**

**Primary test for concordant SNP effects between AD and GIT traits**

| Trait 1 | Trait 2 | Direction | SNP sets ratio | $P_{permuted}$ |
|---|---|---|---|---|
| AD (Jansen et al.[21]) | GERD-UKBB_QSKIN An et al.[23] | + | 144/144 | 0.001[a] |
| Excluding *APOE* and *MHC* regions | PUD (Wu et al.[22]) | + | 139/144 | 0.001[a] |
| | PGM (Wu et al.[22]) | + | 144/144 | 0.001[a] |
| | Gastritis-duodenitis Phecode 535 | + | 144/144 | 0.001[a] |
| | IBS (Wu et al.[22]) | + | 133/144 | 0.001[a] |
| | Diverticulosis Phecode 562 | + | 130/144 | 0.001[a] |
| | IBD (Wu et al.[22]) | + | 42/144 | 0.018[a] |

**Primary test for concordant SNP effects between GIT traits and AD**

| Trait 1 | Trait 2 | Direction | SNP sets ratio | $P_{permuted}$ |
|---|---|---|---|---|
| GERD-UKBB_QSKIN An et al.[23] | AD (Jansen et al.[21]) | + | 144/144 | 0.001[a] |
| PUD (Wu et al.[22]) | Excluding *APOE* and *MHC* regions | + | 138/144 | 0.001[a] |
| PGM (Wu et al.[22]) | | + | 141/144 | 0.001[a] |
| Gastritis-duodenitis Phecode 535 | | + | 135/144 | 0.001[a] |
| IBS (Wu et al.[22]) | | + | 118/144 | 0.001[a] |
| Diverticulosis Phecode 562 | | + | 73/144 | 0.006[a] |
| IBD (Wu et al.[22]) | | + | 19/144 | 0.084 |

*AD* Alzheimer's disease, *GIT* gastro-intestinal tract, *GERD* gastroesophageal reflux disease, *PUD* peptic ulcer disease, *IBS* irritable bowel syndrome, *PGM* GWAS combining disease-diagnosis of PUD and/or GERD and/or medications for their treatments, *IBD* inflammatory bowel disease, *SNP* single-nucleotide polymorphism, *P* P value, *MHC* major histocompatibility complex.
[a]The number of SNP subsets with nominally significant concordant effects is significantly MORE than expected by chance, indicating significant concordance of genetic risk between the pairs of traits.

set 2) including IBD (Table 3). For example, of the total 144 SNP subsets tested with AD as data set 1 (Table 3), all 144 (for GERD, PGM and gastritis-duodenitis), 139 (PUD), 133 (IBS), 130 (diverticulosis) and 42 (IBD) produced Fisher's exact tests with at least nominally significant effect concordance (odds ratio [OR] > 1 and $P < 0.05$). The empirical $P$ values ($P_{permuted}$) for the significant associations, adjusting for the 144 SNP subsets tested (using permutations of 1000 replicates), range from 0.001 to 0.018 (Table 3). These results are significantly more than expected by chance, supporting evidence of genetic overlap between AD and the GIT traits.

By changing the direction of the analysis (in a bidirectional assessment), we tested each of the GIT traits as data set 1 against AD as data set 2 (Table 3). The results indicate evidence of a strong genetic overlap between AD and GERD, PUD, PGM, gastritis-duodenitis, IBS and diverticulosis. The results also suggest (except for IBD) that SNPs that are strongly associated

with AD influence the named GIT traits and vice versa. Overall, findings in SECA are largely consistent with those of LDSC, except in the case of IBD—highlighting how SECA differs from (capacity for a bidirectional assessment) as well as complements LDSC. Notably, and like LDSC, SECA found a significant association between AD and GIT traits with or without the *APOE* region (Table 3 and Supplementary Data 4). Further, replication analyses in SECA produced largely consistent findings as with LDSC (Supplementary Data 5 and 6).

**SNPs and loci shared by AD and GIT disorders**. Leveraging the significant genetic overlap and correlation as well as the substantial GWAS sample sizes, we performed cross-disorder meta-analyses of AD with GERD and PUD. The GWAS for PGM has many cases and overall large sample size (Table 1) and is strongly correlated with GERD ($r_g = 0.99$, $P = 0.000$) and PUD ($r_g = 0.76$,

$P = 4.41 \times 10^{-101}$) [Supplementary Data 7], hence, we also utilised it in a meta-analysis with AD. We aimed at identifying SNPs and loci which were not genome-wide significant in the individual AD or GIT disorder GWAS (i.e., $5 \times 10^{-8} < P_{\text{GWAS-data}} < 0.05$) but reached the status ($P_{\text{meta-analysis}} < 5 \times 10^{-8}$) following a meta-analysis. We additionally identified SNPs and loci which were already established ($P_{\text{GWAS-data}} < 5 \times 10^{-8}$) in AD (Sentinel AD SNPs/loci), but which, following GWAS meta-analyses, were similarly associated with a GIT disorder, and vice versa. Briefly, our GWAS meta-analyses identified shared SNPs and susceptibility loci, some of which are putatively novel for AD or GIT disorders.

First, a meta-analysis of AD and GERD identified a total of 119 SNPs reaching genome-wide significant association ($P_{\text{meta-analysis}} < 5 \times 10^{-8}$, Supplementary Data 8), from which we characterised seven independent ($r^2 < 0.1$) genomic loci—1p31.3, 1q31.1, 3p21.31, 6p21.32, 17q21.32, 17q21.33, 19q13.32 (Table 4). Many SNPs reaching genome-wide significance in these loci were not genome-wide significant in the individual AD and GIT GWAS we analysed but reached the status in the cross-trait meta-analyses (Table 3). Given this premise (that is, $P_{\text{GWAS-data}} > 5 \times 10^{-8} < P_{\text{meta-analysis}}$), the observation that some of the identified loci are known for AD or GIT traits (from other studies) provides support for our cross-trait analysis findings. Specifically, two of the identified loci: (1p31.3 [near *PDE4B*], and 3p21.31 [near *SEMA3F*]) were not previously genome-wide significant for AD (to our knowledge), indicating they are putatively novel for the disorder. Similarly, three of the seven loci: (17q21.32 [*ZNF652*], 17q21.33 [*PHB*], and 19q13.32 [*TOMM40, APOC2, KLC3, ERCC2*]) are putatively novel for GERD given we have no evidence they were previously genome-wide significant for the disorder. A locus at 1q31.1 (near *BRINP3*) was putatively novel for both AD and GERD at the time of our analysis but has now been reported in a recent GERD multi-trait analysis[31]—providing support for our finding. The remaining locus, 6p21.32 (near genes *HLA-DQA2* and *HLA-DRA*) is known for both AD[32] and GIT disorders—IBD[33], ulcerative colitis[34] and Crohn's disease[33]—and now (in our study), GERD.

An additional 175 independent SNPs at 121 loci reached a genome-wide suggestive association ($P_{\text{meta-analysis}} < 1 \times 10^{-5}$, Supplementary Data 9), replicating some of the genome-wide significant loci, including: 1p31.3 (*PDE4B*, lead SNP: rs2840677) and 1q31.1 (*BRINP3*, rs10753964) for AD and GERD. Also, some of the well-established (sentinel) loci for AD in our GWAS showed evidence of association with GERD (Supplementary Data 10) at 8p21.2 (near gene *PTK2B*, and *CHRNA2*, rs28834970). Other AD sentinel loci shared with GERD include: 19q13.32 (near *NECTIN2*, lead SNP: rs12980613), and 19q13.32 (near *KLC3*, rs77988534) [Supplementary Data 10]. Known (sentinel) GERD loci were similarly associated with AD as summarised in Supplementary Data 10.

Second, following a meta-analysis of AD and PUD GWAS, a total of 22 SNPs, at six genomic loci, reached a genome-wide significance ($P_{\text{meta-analysis}} < 5 \times 10^{-8}$, Supplementary Data 11). The identified loci here include 2q37.1, 6p21.32, 8p21.1, 17p13.2, 19q13.32 and 19q13.41 (Table 4). Of the loci found in the AD and GERD meta-analysis, four were replicated in the AD and PUD meta-analysis. Two of these four loci, the 19q13.32 (near *BCL3*, rs28363848), and the 6p21.32 (*HLA-DRA*, rs9270599), were replicated at a genome-wide level of significance. The remaining two loci—*HYAL2*, 3p21.31, $P_{\text{(FE)}} = 5.24 \times 10^{-3}$, rs709210; and *PDE4B*, 1p31.3, $P_{\text{(FE)}} = 2.94 \times 10^{-4}$, rs6695557 (Supplementary Data 12)—were replicated at $7.14 \times 10^{-3}$ level. In addition to the 6p21.32 (*HLA-DRA*, rs9270599), two of the identified loci: at 8p21.1 (near *SCARA3*), and 2q37.1 (near *ATG16L1*) have been reported for AD (*SCARA3*[35], *ATG16L1*[21,32,36]), and GIT traits (*SCARA3*: gastric or stomach ulcer[37], *ATG16L1*: IBD[38], ulcerative

colitis and Crohn's disease[33,39]). Supplementary Data 13 presents 24 independent SNPs, at 21 genomic loci, reaching genome-wide suggestive association ($P_{\text{meta-analysis}} < 1 \times 10^{-5}$) for AD and PUD.

Third, given its large sample size and strong genetic correlation with GERD and PUD, we performed a meta-analysis of PGM with AD thereby identifying 42 SNPs (Supplementary Data 14) at seven independent loci (Table 4) reaching a genome-wide significance level. This analysis replicated, at a genome-wide level ($P_{\text{meta-analysis}} < 5 \times 10^{-8}$), five of the seven genome-wide loci found in the AD and GERD meta-analysis including 1p31.3, 3p21.31, 6p21.32, 17q21.33 and 19q13.32. Additional loci found in the AD and PGM meta-analysis such as 16q22.1 and 1q32.2 were at least genome-wide suggestive ($P_{\text{meta-analysis}} < 1 \times 10^{-5}$) in the AD and GERD analysis, supporting their involvement in the disorders. An additional 23 SNPs, at three loci, were genome-wide suggestive ($P_{\text{meta-analysis}} < 1 \times 10^{-5}$) in the AD and PGM meta-analysis (Supplementary Data 15). Of these, the rs33998678 SNP (16q22.1, *IL34*) is in strong LD ($r^2 = 0.91$) with a genome-wide significant locus found in the AD vs PGM analysis (rs34644948, at 16q22.1, *MTSS2*, Table 4), providing more support for its involvement in AD and GIT traits (GERD and PUD). Similarly, the rs663576 SNP (at 17q21.32, *PHOSPHO1*) is moderately correlated ($r^2 = 0.41$) with a genome-wide significant SNP (rs2584662 at 17q21.33, *PHB*, Table 4), identified in the meta-analysis. This locus (17q21.33) was found in AD and GERD meta-analysis (SNP rs2584662 near *PHB*), supporting its involvement in AD and the GIT traits. Supplementary Data 10 summarises the sentinel AD loci associated with PGM and vice versa.

**Association of identified loci with other traits**. Seven loci reached a genome-wide significance in the meta-analysis of AD and GERD GWAS; most of these loci were replicated in the AD vs PUD and/or AD vs PGM meta-analysis. We queried each of the associated loci for pleiotropic associations with other traits using the GWAS catalogue (https://www.ebi.ac.uk/gwas) and the Open Targets Genetics (https://genetics.opentargets.org) platforms. For three of the loci—1p31.3 (near *PDE4B*), 3p21.31 (near *SEMA3F*), and 1q31.1 (near *BRINP3*)—we have no evidence of their previous association with AD, at a genome-wide level ($P < 5 \times 10^{-8}$). However, and potentially supportive of our findings, the loci have been reported for AD-related phenotypes such as cognitive traits.

For example, *PDE4B* has pleiotropic associations with intelligence[40], educational attainment[41], and sleep-related traits such as insomnia[42]. The locus is also known for other disorders including major depression, stress disorders, schizophrenia, and multiple sclerosis[43]—putative comorbidities of AD[44,45]—among other traits. The loci harbouring *SEMA3F and BRINP3* have similarly been reported for intelligence (*SEMA3F*[46]), general cognitive ability (*SEMA3F*[40]), educational attainment (*SEMA3F*[47], *BRINP3*[41]), insomnia (*SEMA3F* and *BRINP3*[42]) and BMI (*SEMA3F* and *BRINP3*). Sex hormone-binding globulin levels[48] and multi-site chronic pain are some of the traits that have also been linked with *SEMA3F*. Interestingly, BMI, cognitive traits such as intelligence, cognitive performance and even sleep-related traits have been associated with GERD[31]. Taken together, and in further support of their relationship, this observation, suggests that GERD may share genetic links with certain AD-related phenotypes including cognitive and sleep-related traits.

Further, our analysis consistently identified and replicated the 19q13.32 locus (mapped genes: *TOMM40, APOC2, KLC3, ERCC2, BCL3,* and *CD33*) as shared by AD and GIT disorders. While this locus is well known for AD, it has also been linked with GIT traits including IBD[49] (*SYMPK*, lead SNP: rs16980051,

**Table 4 Genome-wide significant independent SNPs and loci for AD and GIT disorders.**

| Independent SNPs | Locus | Lead SNPs | Chr | BP | EA | NEA | $I^2$ | Nearest coding genes/cytoband | Meta-analysis P value | AD P value | GIT disorders P value |
|---|---|---|---|---|---|---|---|---|---|---|---|
| SNPs and loci reaching genome-wide significance after a meta-analysis of AD GWAS and GERD GWAS | | | | | | | | | | | |
| rs12058296 | 1 | rs12058296 | 1 | 66402424 | A | C | 85.75 | PDE4B/1p31.3 | $1.05 \times 10^{-8}$ | $3.22 \times 10^{-5}$ | $1.74 \times 10^{-5}$ |
| rs2503185 | | | 1 | 66461401 | G | A | 91.93 | PDE4B/1p31.3 | $3.44 \times 10^{-8}$ | $9.53 \times 10^{-4}$ | $9.60 \times 10^{-7}$ |
| rs12561863 | 2 | rs12561863 | 1 | 190897608 | A | T | 96.02 | BRINP3/1q31.1 | $1.68 \times 10^{-8}$ | $5.76 \times 10^{-3}$ | $1.05 \times 10^{-7}$ |
| rs3774745 | 3 | rs3774745 | 3 | 50204745 | T | C | 92.74 | SEMA3F/3p21.31 | $2.01 \times 10^{-9}$ | $2.55 \times 10^{-4}$ | $1.64 \times 10^{-4}$ |
| rs28895026 | 4 | rs28895026 | 6 | 32391695 | C | T | 0.00 | HLA-DRA/6p21.32 | $2.06 \times 10^{-8}$ | $1.48 \times 10^{-7}$ | $4.26 \times 10^{-2}$ |
| rs8067459 | 5 | rs2584662 | 17 | 47444113 | C | T | 0.00 | ZNF652/17q21.32 | $3.07 \times 10^{-8}$ | $2.15 \times 10^{-7}$ | $4.48 \times 10^{-2}$ |
| rs2584662 | | | 17 | 47470487 | C | A | 0.00 | PHB/17q21.33 | $7.72 \times 10^{-9}$ | $1.54 \times 10^{-7}$ | $1.02 \times 10^{-2}$ |
| rs11083749 | 6 | rs1132899 | 19 | 45384105 | T | C | 0.00 | TOMM40/19q13.32 | $2.63 \times 10^{-8}$ | $2.46 \times 10^{-7}$ | $3.14 \times 10^{-2}$ |
| rs1132899 | | | 19 | 45448036 | T | C | 94.99 | APOC2/19q13.32 | $1.19 \times 10^{-8}$ | $5.41 \times 10^{-8}$ | $5.53 \times 10^{-3}$ |
| rs117501883 | 7 | rs117501883 | 19 | 45841296 | A | G | 0.00 | KLC3/19q13.32 | $8.96 \times 10^{-9}$ | $7.13 \times 10^{-8}$ | $3.78 \times 10^{-2}$ |
| rs76692930 | | | 19 | 45875851 | T | C | 50.22 | ERCC2/19q13.32 | $3.51 \times 10^{-8}$ | $3.18 \times 10^{-6}$ | $8.32 \times 10^{-4}$ |
| SNPs and loci reaching genome-wide significance after a meta-analysis of AD GWAS and PUD GWAS | | | | | | | | | | | |
| rs36133610 | 1 | rs36133610 | 2 | 234067884 | A | G | 0.00 | ATG16L1/2q37.1 | $1.24 \times 10^{-8}$ | $5.85 \times 10^{-8}$ | $4.90 \times 10^{-2}$ |
| rs9270599 | 2 | rs9270599 | 6 | 32561656 | G | A | 26.29 | HLA-DRA/6p21.32 | $9.12 \times 10^{-9}$ | $5.60 \times 10^{-8}$ | $2.72 \times 10^{-2}$ |
| rs530324 | 3 | rs530324 | 8 | 27491186 | C | G | 76.92 | SCARA3/8p21.1 | $2.27 \times 10^{-8}$ | $3.32 \times 10^{-7}$ | $2.00 \times 10^{-3}$ |
| rs73976310 | 4 | rs73976310 | 17 | 5014212 | A | G | 31.11 | USP6/17p13.2 | $1.20 \times 10^{-8}$ | $7.04 \times 10^{-8}$ | $2.70 \times 10^{-2}$ |
| rs28363848 | 5 | rs28363848 | 19 | 45257201 | T | G | 41.06 | BCL3/19q13.32 | $1.04 \times 10^{-8}$ | $5.63 \times 10^{-8}$ | $2.60 \times 10^{-2}$ |
| rs3852865 | | rs3852865 | 19 | 51714065 | A | G | 58.20 | CD33/19q13.41 | $1.81 \times 10^{-8}$ | $1.63 \times 10^{-7}$ | $9.90 \times 10^{-3}$ |
| rs7245846 | 6 | rs7245846 | 19 | 5173176 | A | G | 5.22 | CD33/19q13.41 | $2.32 \times 10^{-8}$ | $1.19 \times 10^{-7}$ | $4.00 \times 10^{-2}$ |
| SNPs and loci reaching genome-wide significance after a meta-analysis of AD GWAS and PGM GWAS | | | | | | | | | | | |
| rs2840677 | 1 | rs12058296 | 1 | 66333877 | A | T | 66.10 | PDE4B/1p31.3 | $2.43 \times 10^{-8}$ | $5.73 \times 10^{-3}$ | $2.20 \times 10^{-7}$ |
| rs695557 | | | 1 | 66349013 | A | C | 57.16 | PDE4B/1p31.3 | $8.46 \times 10^{-9}$ | $3.02 \times 10^{-3}$ | $1.89 \times 10^{-7}$ |
| rs12058296 | | | 1 | 66402424 | A | C | 84.60 | PDE4B/1p31.3 | $5.02 \times 10^{-9}$ | $3.22 \times 10^{-5}$ | $8.68 \times 10^{-6}$ |
| rs4147104 | 2 | rs4147104 | 1 | 207882194 | A | G | 13.54 | CD46/1q32.2 | $5.47 \times 10^{-9}$ | $1.02 \times 10^{-6}$ | $6.12 \times 10^{-4}$ |
| rs709210 | 3 | rs7642934 | 3 | 50357869 | A | C | 93.99 | HYAL2/3p21.31 | $4.39 \times 10^{-8}$ | $1.55 \times 10^{-2}$ | $6.28 \times 10^{-8}$ |
| rs7642934 | | | 3 | 50174848 | A | G | 93.64 | SEMA3F/3p21.31 | $2.78 \times 10^{-8}$ | $7.47 \times 10^{-3}$ | $8.19 \times 10^{-8}$ |
| rs2858331 | 4 | rs2858331 | 6 | 32681277 | G | A | 61.35 | HLA-DQA2/6p21.32 | $3.08 \times 10^{-10}$ | $1.24 \times 10^{-7}$ | $1.18 \times 10^{-4}$ |
| rs28895026 | | | 6 | 32391695 | C | T | 0.00 | HLA-DRA/6p21.32 | $5.43 \times 10^{-9}$ | $1.48 \times 10^{-7}$ | $8.38 \times 10^{-3}$ |
| rs34644948 | 5 | rs34644948 | 16 | 70681658 | T | C | 0.00 | MTSS2/16q22.1 | $2.11 \times 10^{-8}$ | $1.98 \times 10^{-7}$ | $3.13 \times 10^{-2}$ |
| rs2584662 | 6 | rs2584662 | 17 | 47470487 | C | A | 0.00 | PHB/17q21.33 | $3.94 \times 10^{-9}$ | $1.98 \times 10^{-7}$ | $4.91 \times 10^{-3}$ |
| rs11083749 | 7 | rs11083749 | 19 | 45384105 | T | C | 0.00 | TOMM40/19q13.32 | $2.84 \times 10^{-8}$ | $1.98 \times 10^{-7}$ | $3.45 \times 10^{-2}$ |

Meta-analysis model use was the RE2. RE2: GWAS meta-analysis method that adjusts for SNP effects heterogeneity.
SNP single-nucleotide polymorphism, Chr chromosome, EA effect allele, NEA non-effect allele, $I^2$ $I$-square for heterogeneity assessment, Se standard error, P P value.

GRCh37: 19:46,345,886), and gut microbiota[50], thus, highlighting an association of AD with not only GIT disorders, but also the gut microbiome. This premise is important given previous evidence of genetic links between dysbiosis, neurological (AD, for instance) and GIT disorders[15,22,51,52], and may underscore the need for a renewed focus on the genetics of gut-brain connection (including the gut microbiome) to better understand the underlying mechanisms of AD. Similar to other identified loci, the 19q13.32 locus also displays pleiotropic association with many AD-related phenotypes: intelligence[53], cognitive impairment test score[54], t-tau and beta-amyloid 1–42 measurements, hippocampal atrophy rate, memory performance, and educational attainment[41]. Supplementary Data 16 and 17 summarise other traits previously reported for loci at 6p21.32 (near *HLA-DRA*) and 17q21.32 (near *ZNF652* and *PHB*).

**Shared genomic regions identified in GWAS-PW analysis.** Using a colocalization analysis in GWAS-PW[27], we assessed shared genomic regions between AD and each of GERD and PGM (Supplementary Data 18). The results of this analysis confirm all the loci identified in the meta-analyses (except on chromosome 3) are shared by AD and the respective GIT traits (model 4 posterior probability [PPA 4] > 0.9, Supplementary Data 18). While the findings also suggest that the causal variants might be different (PPA 3 < 0.5), we note that when variants in a locus are in strong LD, which may be the case in this study, GWAS-PW is limited in its ability to correctly distinguish model 3 (PPA 3) from model 4 (PPA 4)[27]. Additional shared genomic regions, in chromosomes 1, 6, 16, 17 and 19 having PPA 4 > 0.90 were identified for AD and the GIT traits (Supplementary Data 18). Also, we identified a locus on chromosome 17, having PPA 3 > 0.80, and implicating the SNP rs2526380 (17q22, near *TSPOAP1*) in both AD and GERD. The posterior probability that this SNP is a causal variant for both AD and GERD under model 3[27] is high at 0.99 (Supplementary Data 18).

**Results of causal association analysis between AD and GIT disorders.** We assessed the potential causal relationship between AD (as the outcome variable) and GERD (as the exposure variable) using the two-sample MR method. We found no evidence of a causal relationship between AD and GERD, irrespective of the direction of the analysis (AD or GERD as the outcome or exposure variable) [Table 5]. For sensitivity testing, we implemented three additional models of MR analysis—MR-Egger, weighted median, and the MR-PRESSO (Mendelian Randomization Pleiotropy RESidual Sum and Outlier). Results from these methods agree with those of the Inverse Variance Weighted (IVW) model supporting a lack of evidence for a causal association between AD and GERD (Table 5 and Supplementary Data 19). We carried out further MR analysis assessing AD against each of PUD, PGM, IBS, diverticular disease, and IBD, and vice versa. Findings similarly reveal no evidence for a causal relationship between AD and each of the GIT disorders assessed (Supplementary Data 19).

We also used the Latent Causal Variable (LCV) approach[29] to test for a causal relationship between AD and each of the GIT disorders. The results of LCV suggest a partial causal influence of gastritis-duodenitis (genetic causal proportion [GCP] = −0.69, $P = 0.0026$), on AD (Table 6). The result was in the reverse direction for diverticular disease (GCP = 0.23, $P = 0.000272$), suggesting AD may partially cause diverticular disease. Using another set of GWAS (Table 6), we tested the reproducibility of the partial causal association results for gastritis-duodenitis and diverticular disease, neither of which was reproduced, hence, the need for the findings to be further assessed in future studies.

**Table 5 Summary of MR analysis results for AD and GIT disorders.**

| Exposure (nSNPs) | Outcome | IVW | | Weighted median | | MR-Egger | | MR-PRESSO | | | | | MR-Egger Intercept | |
|---|---|---|---|---|---|---|---|---|---|---|---|---|---|---|
| | | Beta | P | Beta | P | Beta | P | Global test P | Raw beta | P | Corrected beta | P | Intercept | P |
| AD (28) | GERD | −0.053 | 0.266 | 0.011 | 0.860 | −0.059 | 0.362 | 0.113 | −0.052 | 0.276 | — | — | 0.00034 | 0.879 |
| GERD (24) | AD | 0.014 | 0.351 | −0.002 | 0.920 | −0.053 | 0.597 | 0.435 | 0.0136 | 0.361 | — | — | 0.0025 | 0.502 |
| AD (28) | PUD | 0.036 | 0.651 | 0.144 | 0.211 | 0.071 | 0.504 | 0.113 | 0.036 | 0.60 | — | — | −0.002 | 0.612 |
| PUD (8) | AD | 0.021 | 0.238 | 0.025 | 0.122 | 0.055 | 0.658 | 0.0104 | 0.021 | 0.277 | 0.0331 | 0.053 | −0.00291 | 0.770 |
| AD (28) | PGM | −0.061 | 0.112 | −0.016 | 0.769 | −0.045 | 0.391 | 0.231 | −0.061 | 0.123 | — | — | −0.001 | 0.631 |
| PGM (17) | AD | 0.023 | 0.322 | −0.005 | 0.837 | −0.148 | 0.199 | 0.017 | 0.023 | 0.337 | 0.009 | 0.661 | 0.007 | 0.133 |
| AD (28) | Gastritis-D[a] | −0.085 | 0.267 | −0.101 | 0.273 | −0.173 | 0.098 | 0.034 | −0.085 | 0.277 | — | — | 0.0046 | 0.196 |
| AD (28) | IBS[a] | 0.043 | 0.623 | 0.142 | 0.123 | −0.016 | 0.888 | 0.0012 | 0.043 | 0.626 | 0.010 | 0.892 | 0.0032 | 0.438 |
| AD (28) | Diverticular | −0.055 | 0.597 | −0.214 | 0.105 | −0.12 | 0.397 | 0.094 | −0.055 | 0.601 | — | — | 0.0034 | 0.483 |
| Diverticular (16) | AD | −0.001 | 0.883 | −0.001 | 0.905 | 0.007 | 0.811 | 0.316 | −0.001 | 0.884 | — | — | −0.00076 | 0.773 |
| AD (28) | IBD | 0.254 | 0.104 | 0.365 | 0.094 | 0.277 | 0.231 | 0.327 | 0.254 | 0.115 | — | — | −0.00097 | 0.885 |
| IBD (24) | AD | −0.0005 | 0.895 | −0.003 | 0.526 | 0.004 | 0.607 | 0.316 | −0.0005 | 0.895 | — | — | 0.00073 | 0.497 |

*nSNP* number of SNPs utilised as instrumental variables, *SNP* single-nucleotide polymorphism, *AD* Alzheimer's disease, *GERD* gastroesophageal reflux disease, *PUD* peptic ulcer disease, *PGM* GWAS combining disease-diagnosis of PUD and/or GERD and/or medications for their treatments, *Diverticular* diverticular disease, *IBS* irritable bowel syndrome, *IBD* inflammatory bowel disease, *IVW* inverse variance weighted, *P* P value, *MR-PRESSO* Mendelian Randomization Pleiotropy RESidual Sum and Outlier, *Gastritis-D* gastritis-duodenitis. Note spaces marked with a dash indicate that there were no outlier SNPs and hence there was no outlier corrected results in the MR-PRESSO analysis.
[a]Only one genome-wide significant SNPs available for IBS, and 3 for Gastritis-D, so we are unable to carry out MR using the traits as the exposure variable.

Conversely, we found a significant association between AD and lansoprazole use (GCP = −0.38, P = 0.001129).

**Gene-based association analysis.** Using SNPs that overlapped AD and GERD GWAS, we performed gene-based analyses in MAGMA (implemented in the FUMA[55] platform), thereby identifying a total of 18,929 protein-coding genes for each of the traits. Applying a threshold P-value of $2.64 \times 10^{-6}$ (0.05/18929—Bonferroni correction for testing 18,929 genes), we identified 64 genome-wide significant ($P_{gene} < 2.64 \times 10^{-6}$) genes for AD (Supplementary Data 20), 44 for GERD (Supplementary Data 21) and 75 for PGM (Supplementary Data 22). Using the Fisher's Combined P-value (FCP) method, a total of 46 genome-wide significant ($P_{FCP} < 2.64 \times 10^{-6}$) genes shared by AD and GERD were identified (Supplementary Data 23), 10 of which were not previously significant in our AD or GERD GWAS, at the $P_{gene} < 2.64 \times 10^{-6}$ threshold, adjusting for multiple testing (Table 7), but are in known AD or GIT trait loci. It is noteworthy

that some of the identified AD and GERD shared genes are in chromosomal locations found in our meta-analysis, including 1p31.3 (*PDE4B*), 3p21.31, (*SEMA3F, HYAL2, IP6K1*), 6p21.32 (*HLA-DRA*) and 19q13.32 (Supplementary Data 23). Combining P-values by weighting based on sample size (the weighted Stouffer's method) produced a similar pattern of results (as the FCP) for AD and GERD (Supplementary Data 24). We also replicated a similar pattern of findings in gene-based analysis (and FCP) using the AD and the PGM GWAS (Table 7 Supplementary Data 25).

**Biological pathways and mechanisms shared by AD and GIT disorders.** We performed pathway-based functional enrichment analyses in the g: Profiler platform[56] to functionally interpret genes overlapping AD and GIT disorders and gain biological insight from their commonalities. First, we investigated genes overlapping AD and GERD (at $P_{gene} < 0.05$, FCP < 0.02) and identified several biological pathways that were overrepresented (Fig. 2 and Supplementary Data 26), implying they have a role in the mechanisms underlying both AD and GERD. Pathways related to membrane trafficking and metabolism, alteration, lowering or inhibition of lipids were significantly enriched (Supplementary Data 26). These included plasma lipoprotein assembly, remodelling, and clearance ($P_{adjusted} = 2.01 \times 10^{-3}$), cholesterol metabolism ($P_{adjusted} = 4.99 \times 10^{-2}$), plasma lipoprotein assembly ($P_{adjusted} = 3.45 \times 10^{-5}$), and triglyceride-rich plasma lipoprotein particle ($P_{adjusted} = 5.23 \times 10^{-9}$), among others. Also, lipase inhibitors ($P_{adjusted} = 6.08 \times 10^{-3}$) and the statin (3-hydroxy-3-methylglutaryl-coenzyme A reductase inhibitors) pathway ($P_{adjusted} = 3.99 \times 10^{-2}$) were significantly enriched for AD and GERD (Supplementary Data 27), suggesting mechanisms of these medications may find therapeutic application in AD and GIT disorders.

Pathways related to the immune system were also over-represented for both AD and GERD as evidenced by the identification of immune or autoimmune-related disorders such

**Table 6 Partial causality assessment using the Latent Causal Variable approach.**

| Trait 1 | Trait 2 | GCP | SE | P |
|---|---|---|---|---|
| AD | GERD | −0.01 | 0.58 | 0.64 |
| | PUD | 0.49 | 0.32 | 0.24 |
| | PGM | −0.45 | 0.37 | 0.22 |
| | Gastritis-duodenitis (Main ICD10: K29) | −0.69 | 0.27 | 0.0026 |
| | IBS | 0.35 | 0.29 | 0.38 |
| | Diverticular disease (Main ICD10: K57) | 0.23 | 0.10 | 0.000272 |
| | Lansoprazole | −0.38 | 0.17 | 0.001129 |

*AD* Alzheimer's disease, *GCP* genetic causal proportion, *SE* standard error, *P* P value, *GERD* gastroesophageal reflux disease, *PUD* peptic ulcer disease, *PGM* GWAS combining disease-diagnosis of PUD and/or GERD and/or medications for their treatments.

**Table 7 Shared genes reaching genome-wide significance for AD and GIT traits.**

| SYMBOL | CHR | CYTOBAND | START | STOP | P-AD | P-GIT | FCP-value |
|---|---|---|---|---|---|---|---|
| Genome-wide significant genes shared by AD and GERD | | | | | | | |
| MON1A | 3 | 3p21.31 | 49946302 | 49967606 | 2.42E−02 | 3.14E−06 | 1.33E−06 |
| IP6K1 | 3 | 3p21.31 | 49761727 | 49823975 | 1.47E−02 | 1.05E−05 | 2.57E−06 |
| HLA-DRA | 6 | 6p21.32 | 32407619 | 32412823 | 1.38E−05 | 7.06E−06 | 2.34E−09 |
| PGBD1 | 6 | 6p22.1 | 28249314 | 28270326 | 9.62E−03 | 4.08E−06 | 7.08E−07 |
| NKAPL | 6 | 6p22.1 | 28227098 | 28228736 | 1.33E−03 | 4.64E−05 | 1.09E−06 |
| ZKSCAN8 | 6 | 6p21 | 28109688 | 28127250 | 2.11E−02 | 3.92E−06 | 1.43E−06 |
| C6orf10 | 6 | 6p21.32 | 32256303 | 32339684 | 2.96E−05 | 4.20E−03 | 2.10E−06 |
| TMEM106B | 7 | 7p21.3 | 12250867 | 12282993 | 1.33E−04 | 1.04E−03 | 2.32E−06 |
| ZNF689 | 16 | 16p11.2 | 30613879 | 30635333 | 9.71E−06 | 1.55E−02 | 2.51E−06 |
| FOXA3 | 19 | 19q13.32 | 46367247 | 46377055 | 6.31E−04 | 5.45E−05 | 6.25E−07 |
| Genome-wide significant genes shared by AD and PGM | | | | | | | |
| CR1L | 1 | 1q32.2 | 207818458 | 207911761 | 2.70E−05 | 4.47E−03 | 2.05E−06 |
| HYAL2 | 3 | 3p21.31 | 50355221 | 50360337 | 3.63E−02 | 3.49E−06 | 2.14E−06 |
| C6orf10 | 6 | 6p21.32 | 32256303 | 32339684 | 2.32E−05 | 3.64E−04 | 1.65E−07 |
| NKAPL | 6 | 6p22.1 | 28227098 | 28228736 | 1.33E−03 | 7.31E−06 | 1.89E−07 |
| CLIC1 | 6 | 6p21.33 | 31698358 | 31707540 | 7.64E−04 | 1.88E−05 | 2.74E−07 |
| ZSCAN9 | 6 | 6p22.1 | 28192664 | 28201260 | 1.92E−03 | 1.46E−05 | 5.17E−07 |
| TRIM39-RPP21 | 6 | 6p22.1 | 30297359 | 30314631 | 9.07E−03 | 3.63E−06 | 6.01E−07 |
| ZSCAN12 | 6 | 6p22.1 | 28346732 | 28367511 | 1.10E−02 | 7.36E−06 | 1.40E−06 |
| BAG6 | 6 | 6p21.33 | 31606805 | 31620482 | 6.68E−03 | 1.61E−05 | 1.84E−06 |
| FBXO46 | 19 | 19q13.32 | 46213887 | 46234162 | 7.62E−04 | 4.19E−05 | 5.83E−07 |
| RSPH6A | 19 | 19q13.32 | 46298968 | 46318577 | 2.76E−04 | 1.16E−04 | 5.85E−07 |

Note: genes reported in this Table were not previously genome-wide in the gene-based analysis for the individual AD and GIT GWAS analysed but reached the status following FCP analysis.
*CHR* chromosome, *P-AD* P value for Alzheimer's disease, *P-GIT* P value for gastrointestinal tract trait, *FCP* Fisher's combined P value, *GERD* gastroesophageal reflux disease, *PGM* GWAS combining disease-diagnosis of PUD and/or GERD and/or medications for their treatments.

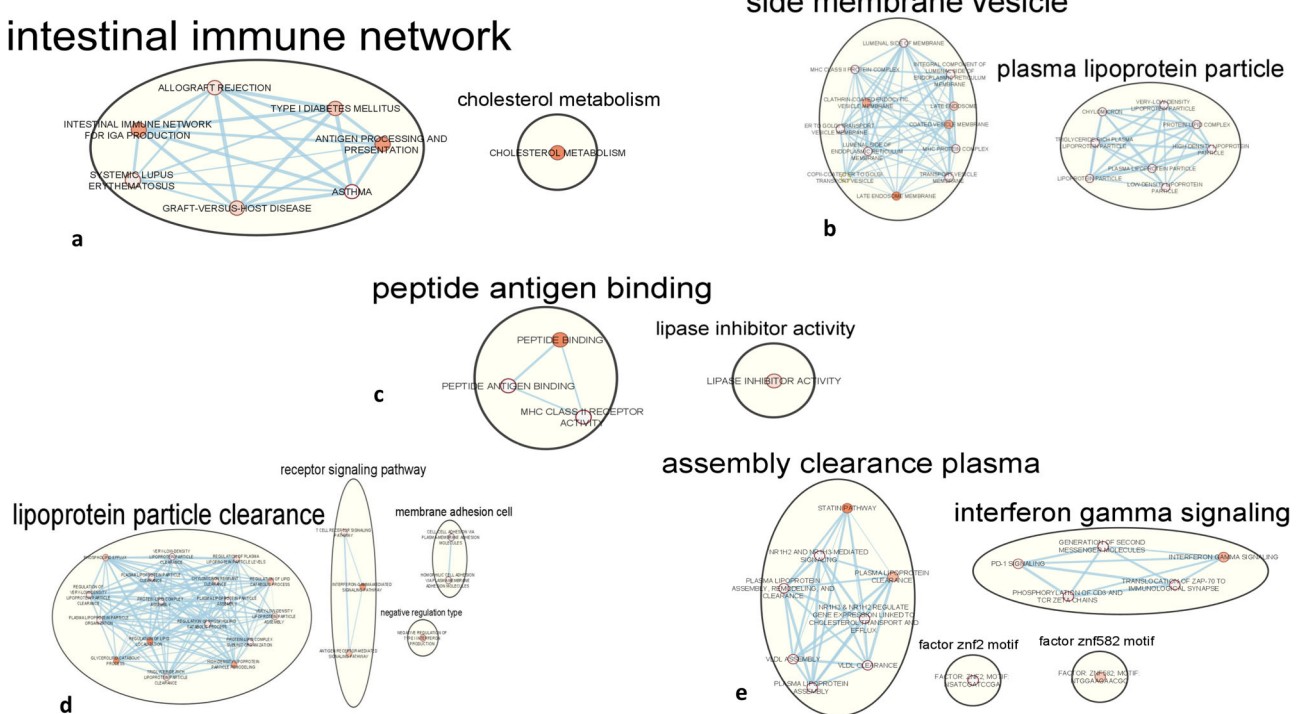

**Fig. 2 Clusters of significantly enriched biological pathways for AD and GERD. a** KEGG: Kyoto Encyclopedia of Genes and Genomes pathways: *intestinal immune network* (allograft rejection, intestinal immune network for IGA production, type 1 diabetes mellitus, systemic lupus erythematous, antigen processing and presentation, graft-versus-host disease, asthma), and *cholesterol metabolism* (cholesterol metabolism). **b** Gene Ontology: Cellular Components: *side membrane vesicle* (lumenal side of membrane, MHC class II protein complex, integral component of lumenal side of endoplasmic reticulum [ER] membrane, clathrin-coated endocytic vesicle membrane, late endosome, ER to Golgi transport vesicle membrane, coated vesicle membrane, lumenal side of ER membrane, MHC protein complex, COPII-coated ER to Golgi transport vesicle, transport vesicle membrane, late endosome membrane), and *plasma lipoprotein particle* (chylomicron, very low-density lipoprotein [VLDL] particle, triglyceride-rich plasma lipoprotein particle, plasma lipoprotein particle, lipoprotein particle, LDL lipoprotein particle). **c** Gene Ontology: Molecular Function: peptide antigen binding (peptide binding, peptide antigen binding, MHC class II receptor activity) and *lipase inhibitor activity* (lipase inhibitor activity). **d** Gene Ontology: Biological Pathway: *lipoprotein particle clearance* (phospholipid efflux, VLDL particle clearance, regulation of plasma lipoprotein particle levels, plasma lipoprotein particle clearance, chylomicron remnant clearance, regulation of lipid catabolic process, regulation of VLDL particle clearance, protein-lipid complex assembly, plasma lipoprotein particle organisation, regulation of phospholipid catabolic process, VLDL particle assembly, regulation of lipid localisation, glycolipid catabolic process, triglyceride-rich lipoprotein particle clearance, high density lipoprotein particle remodelling), *receptor signalling pathway* (T cell receptor signalling pathway, interferon-gamma-mediated signalling pathway, antigen receptor-mediated signalling pathway), *membrane adhesion cell* (cell-cell adhesion via plasma membrane adhesion molecules, homophilic cell adhesion via plasma membrane adhesion molecules), and *negative regulation type* (negative regulation of type I interferon production). **e** Reactome, Wiki pathway and Transcription Factor Binding site: *assembly clearance plasma* (statin pathway, NR1H2 and NR1H3-mediated signalling, plasma lipoprotein assembly, remodelling, and clearance, plasma lipoprotein clearance, NR1H3 and NR1H2 regulated gene expression linked to cholesterol transport and efflux, VLDL assembly, VLDL clearance, plasma lipoprotein assembly), *interferon-gamma signalling* (PD-1 signalling, generation of second messenger molecules, interferon-gamma signalling phosphorylation of CD3 and TCR ZETA chains, translocation of ZAP-70 to Immunological synapse), *Factor*: ZNF2 motif, and ZNF582 motif. Supplementary Data 26 provides additional details about these biological pathways. AD Alzheimer's disease, GERD gastroesophageal reflux disease.

as asthma ($P_{adjusted} = 3.53 \times 10^{-3}$), systemic lupus erythematosus ($P_{adjusted} = 7.88 \times 10^{-3}$), and type I diabetes mellitus ($P_{adjusted} = 2.47 \times 10^{-2}$). Other immune-related pathways identified include the intestinal immune network for IgA production ($P_{adjusted} = 4.07 \times 10^{-2}$), programmed cell death protein 1 (PD-1) signalling ($P_{adjusted} = 5.24 \times 10^{-3}$), translocation of ZAP-70 to immunological synapse ($P_{adjusted} = 2.44 \times 10^{-3}$) and interferon-gamma signalling pathways ($P_{adjusted} = 2.45 \times 10^{-2}$) [Supplementary Data 26].

Following enrichment mapping and auto-annotation, the identified biological pathways were clustered into six themes of biological mechanisms, namely: 'lipoprotein particle clearance,' 'receptor signalling pathway,' 'side membrane vesicle and cell adhesion,' 'peptide antigen binding,' 'intestinal immune network,' and 'interferon-gamma signalling' (Fig. 2). Moreover, a pathway-based analysis using genes that overlapped AD and PGM GWAS

(at $P_{gene} < 0.05$) replicated some of the pathways identified for AD and GERD, including 'plasma lipoprotein assembly, remodelling, and clearance' ($P_{adjusted} = 3.01 \times 10^{-4}$), 'peptide antigen binding' ($P_{adjusted} = 2.28 \times 10^{-3}$), and 'triglyceride-rich plasma lipoprotein particle' ($P_{adjusted} = 6.60 \times 10^{-8}$) [Supplementary Data 27]. Also, we performed pathway-based analysis separately for GERD and AD GWAS, the full results of which are presented in Supplementary Data 28 and 29, respectively.

## Discussion

We present the first comprehensive assessment (to the best of our knowledge) of the shared genetics of AD with GIT disorders by analysing large-scale GWAS summary data using multiple statistical genetic approaches. Consistent with previous conventional observational studies[3–9], our findings confirm a risk-increasing

relationship between AD and GIT disorders and provide insights into their underlying biological mechanisms. In contrast to the positive genetic correlation between AD and other GIT disorders, LDSC found no significant genetic correlation between AD and IBD, which may be due to the relatively small number of cases and sample size of the IBD GWAS. Based on the effective sample size estimates, the IBD GWAS is underpowered compared to other GIT data sets. Supporting this premise, SECA revealed a significant association between AD (as data set 1) against IBD (as data set 2), but not the other way around. The AD GWAS has a larger sample size, providing a more robust association on which to condition (select independent) SNPs for concordance analysis which may explain why the significant association was not bi-directional unlike the case for other GIT traits. Future studies, nonetheless, need to confirm this relationship, as more powerful IBD GWAS becomes available.

Evidence of significant genetic overlap and correlation reflects not only shared genetic aetiologies (biological pleiotropy) but also suggests a possible causal association between AD and the GIT traits (vertical pleiotropy). Using LCV, we detected a partial causal association between AD and gastritis-duodenitis, lansoprazole, and diverticular disease. However, this partial causal association was not evident in reproducibility testing. The inconclusive LCV findings should be cautiously interpreted, and a reassessment of the results, in future studies, is warranted. Conversely, all MR analyses provided no evidence for a significant causal relationship between AD and GIT traits, indicating that shared genetics and common biological pathways may best explain the association between AD and these GIT disorders.

We performed GWAS meta-analysis, thereby identifying seven shared loci reaching genome-wide significance for AD and GERD. The loci, including 1p31.3 (PDE4B), 3p21.31 (SEMA3F, HYAL2), 6p21.32 (HLA-DRA), and 19q13.32 (TOMM40, APOC2, ERCC2, BCL3, and KLC3), were replicated in AD vs PUD and AD vs PGM meta-analyses and largely reinforced in colocalisation (GWAS-PW) as well as gene-based association analyses. Notably, the independent SNP rs12058296 (1p31.3), mapped to the phosphodiesterase 4B (PDE4B) gene. Inhibition of PDE4B (or its subtypes) has shown promise for inflammatory diseases[57–60]. Indeed, recent evidence supports the potent anti-inflammatory, pro-cognitive, neuro-regenerative, and memory-enhancing properties of PDE4 inhibitors (PDE4B, in particular[61]), making them plausible therapeutic targets for AD[59,60] and GIT disorders[58]. Other identified independent genome-wide significant SNPs and loci mapped to genes including CD46, SEMA3F, HLA-DRA, MTSS2, PHB, and TOMM40. The CD46 gene is a complement regulator which is bactericidal to Helicobacter (H) pylori[62] and was also recently identified for AD in a transcriptome analysis[63], making it a plausible candidate in both AD and GIT disorders.

We identified biological pathways, significantly enriched for genes overlapping AD and GIT disorder (GERD, and PUD) GWAS in pathway-based analyses. Notably, lipid-related, and autoimmune pathways were overrepresented. There is a close link between autoimmunity and lipid abnormalities[64], and consistent with previous studies[65–69], our findings highlight the importance of lipids homoeostasis in AD and GIT traits. In AD, for example, hypercholesterolaemia is believed to increase the permeability of the blood-brain barrier system, facilitating the entry of peripheral cholesterol into the CNS, and resulting in abnormal cholesterol metabolism in the brain[65,66]. Amyloidogenesis, alteration of the amyloid precursor protein degradation, accumulation of Aβ, and subsequent cognitive impairment have all been linked with elevated cholesterol in the brain[66,70–72]. Similarly, while the exact roles of lipids in GIT disorders are unclear, H. pylori is believed to cause or worsen abnormal serum lipid profiles through chronic inflammatory processes, and eradication of the infection enhances lipid homoeostasis[68,69].

The mechanisms of association between AD and lipid dysregulation relate to the 'gut–brain axis', alterations in GIT microbiota and the immune system[10,66]. Moreover, lipid dysregulation is central to the interplay of AD, gut microbiota, and GIT disorders[10,66], thus, suggesting the therapeutic potential of lipid-lowering medications such as lipase inhibitors and statins (identified in our study) in AD and GIT disorders. Lipase inhibitors (orlistat) prevent intestinal dietary lipid absorption, and lower total plasma triglycerides and cholesterol levels[73,74], making them a preferred pharmacological treatment for obesity[73]. The connection between AD, lipid dysregulation, dysbiosis and the 'gut-brain axis'[10,66], may, thus, support the potential utility of lipase inhibitors in AD. Lipases, including monoacylglycerol, diacylglycerol, and lipoprotein lipases are involved in AD pathology, and can also effectively be inhibited by orlistat[74]. Similarly, statins possess anti-inflammatory, immune-modulating and gastroprotective properties[75,76], and their active use significantly reduced PUD risk[76] as well as enhanced H. pylori eradication[77]. Statins also improve cognitive ability and reduce neurodegeneration risks, making them potentially beneficial in AD[78,79]. However, there is evidence suggesting a paradoxical predisposition to reversible dementia for statins[78,79]. While this finding has been challenged[78], it may highlight a need to identify AD patients for whom statins will be beneficial, consistent with the model of personalised health.

Our findings have implications for practice and further studies. First, results highlighting lipid-related mechanisms support the roles of abnormal lipid profiles in the aetiologies of the disorders, which may be potential biomarkers for AD and GIT disorders (or their comorbidity). Second, our findings underscore the importance of lipid homoeostasis. The dietary approach is one effective preventive as well as non-pharmacologic approach for the management of hyperlipidaemia, and overall, this is consistent with findings in this study. Indeed, adherence to a 'Mediterranean' diet (low in lipids) is recognised as beneficial both in AD[80] and GIT disorders[81]. Thus, a recommendation for healthy diets, early in life, may form part of the lifestyle modifications for preventing AD and GIT disorders. The clinical utility of these recommendations will need to be further investigated and validated. Third, our study identifies lipase inhibitors and statin pathways in the mechanisms of AD and GIT disorders, which may be a potential therapeutic avenue to explore in the disorders. We hypothesise that individuals with comorbid AD and GIT traits may gain benefits from these therapies. There is a need to test this hypothesis using appropriate study designs including randomised control trials. Fourth, our study implicates the PDE4B, and given the evidence in the literature[58–61], we propose that treatment targeted at its inhibition may be promising in comorbid AD and GIT traits. Lastly, while our findings do not necessarily indicate that AD and GIT disorders will always co-occur, they support their shared biology; thus, early detection of AD may benefit from probing impaired cognition in GIT disorders.

The use of multiple, complementary statistical genetic approaches enables a comprehensive analysis of the genetic associations between AD and GIT disorders and is a major strength of this study. Also, we analysed well-powered GWAS data, meaning our findings are generally not affected by small sample size, possible reverse causality, or confounders that conventional observational studies often suffer from. Nonetheless, our study has limitations that should be considered alongside the present findings. First, the GWAS for AD combined clinically diagnosed cases of AD with proxies (AD-by-proxy—individuals whose parents were diagnosed with AD). Given the high

correlation between the GWAS with and without the 'AD-by-proxy' cases[21], we argue as did others[21] that combining them is valid, especially for sample size improvement, which is critical to ensuring adequately powered GWAS analysis. Second, analyses were restricted to participants of mainly European ancestry in our study, thus, findings may not be generalisable to other ancestries. Third, GIT traits GWAS were combinations of several data sources: primary care, hospital admission, medication use, and self-reported records. While there is a potential for misdiagnosis or accuracy of self-reported data, their use is well justified given the correlation in effect sizes of the data with other sources[22]. Moreover, additional data from other sources including ICD-10 were utilised with consistent results across these GWAS.

In conclusion, this study provides genetic insights into the long-standing debate and the observed relationship of AD with GIT disorders, implicating shared genetic susceptibility. Our findings support a significant risk increasing (but non-causal) genetic association between AD and GIT traits (GERD, PUD, PGM, gastritis-duodenitis, IBS, and diverticular disease). Also, we identified genomic regions and genes, shared by AD and GIT disorders that may potentially be targeted for further investigation, particularly, the *PDE4B* gene (or its subtypes) which has shown promise in inflammatory diseases[57–60]. Our study also underscores the importance of lipid homoeostasis and the potential relevance of statins and lipase inhibitors in AD, GIT disorders or their comorbidity. To our knowledge, this is the first comprehensive study to assess these relationships using statistical genetic approaches. Overall, these findings advance our understanding of the genetic architecture of AD, GIT disorders, and their observed co-occurring relationship.

## Methods

**GWAS summary statistics.** The GWAS data utilised in the present study are summarised in Table 1 with further cohort-specific details, including effective sample size estimates, provided in Supplementary Data 1. The data were sourced from popular GWAS databases, repositories, and large research consortia/groups. The GWAS summary data for 'clinically diagnosed AD and AD-by-proxy'[21] (the largest publicly available AD GWAS) was used as our AD GWAS data. This GWAS has large sample size (cases = 71,880, controls = 383,378, sample size [N] = 455,258) and, thus, increased power for detecting genetic variants of small to moderate effect sizes. More specific details about the data have been published[21]. GIT traits including PUD (cases = 16,666, controls = 439,661, N = 456,327), IBS (cases = 28,518, controls = 426,803, N = 455,321), and IBD (cases = 7045, controls = 449,282, N = 456,327) were assessed against AD. The GWAS for the traits were obtained from the recently published GIT GWAS[22] and other sources located through the GWAS Atlas[24] (Supplementary Data 1). Clinically, PUD medications are indicated in GERD and gastritis, accordingly, GWAS combining diagnosis for PUD and/or GERD and/or medications commonly used for these disorders (PGM) have been conducted[22], potentially identifying people with PUD or GERD. This GWAS has a large sample size (cases = 90,175, controls = 366,152, N = 456,327), and as was the case in the original publication[22], we utilised the data for analysis in the present study, as a proxy for PUD or GERD. These GIT GWAS were well characterised and, where possible, validated as described in the original publication[22].

Additionally, we utilised a well-characterised GWAS for GERD (cases = 71,522, controls = 261,079, N = 332,601), which combined data sets from the UK Biobank and the QSKIN study[23]. Gastritis-duodenitis (cases = 28,941, controls = 378,124, N = 407,065) and diverticular disease (cases = 27,311, controls = 334,783, N = 362,094) GWAS from the Lee Lab (https://www.leelabsg.org/resources) were also used in this study. We utilised additional (available) GWAS summary data (Table 1 and Supplementary Data 1) sourced from public repositories used for possible replication of our genetic overlap and correlation (LDSC and SECA) findings. A comprehensive description of the quality control procedures for each of the GWAS data and their analysis are available through the corresponding publications (Table 1 and Supplementary Data 1). Our preliminary analysis indicates that there is no significant sample overlap between the AD GWAS and each of the GIT GWAS assessed in this study (Supplementary Data 2), ruling out the possibility of bias from such occurrence.

**Linkage disequilibrium score regression analysis (LDSC).** We assessed and quantified SNP-level genetic correlation between AD and GIT disorders using the LDSC[25] analysis method (https://github.com/bulik/ldsc/wiki/Heritability-and-Genetic-Correlation). LDSC assesses and distinguishes the contributions of

polygenicity, sample overlaps, and population stratification to the heritability and genetic correlation between traits[25]. In the present study, we performed LDSC analysis using the standalone version of the software and by following the procedures provided by the program developer (https://github.com/bulik/ldsc). The apolipoprotein E (*APOE*) region has a large effect on the risk of AD; hence, we excluded *APOE* and the 500 kilobase (kb) flanking region (hg19, 19:44,909,039–45,912,650) from the AD GWAS for this analysis. We also excluded SNPs in the 26–36 megabase region of chromosome six from the data given the complex LD structure in the human major histocompatibility complex (MHC). To assess possible sample overlap between AD GWAS and each of the GIT GWAS, we performed LDSC correlation analysis with the genetic covariance intercept unconstrained. The result of this analysis indicates that the estimated genetic covariance intercepts were not significantly different from zero (Supplementary Data 2), indicating no significant sample overlap between our AD and GIT GWAS. Thus, we constrained the intercept in the reported genetic correlation analysis. We applied Bonferroni adjustment for testing the effects of seven GIT traits on AD $(0.05/7 = 7.1 \times 10^{-3})$, and all genetic correlation results surviving this adjustment were considered significant while those having $P < 0.05$ were regarded as nominally significant.

**SNP effect concordance analysis (SECA).** We used the standalone version of the SECA software pipeline to perform SNP-level genetic overlap assessment and statistical tests between AD and GIT disorders. A detailed description of the SECA software and methods has been published[26]. Briefly, SECA accepts a pair of GWAS (data set 1 and data set 2) as input and performs a range of analyses to assess concordance in effect direction between a pair of traits—AD and GIT disorders in the present study. First, we carried out quality control to exclude all non-rsID(s) and duplicate variants in data set 1 and align SNP effects to the same effect allele across data set 1 and data set 2. Second we performed two rounds of P-value informed LD clumping in data set 1 (first clumping: -clump-r² 0.1, -clump-kb 1000; second clumping: -clump-r² 0.1, -clump-kb 10000) using PLINK 1.90[30].

Third, SECA partitions independent SNPs resulting from LD clumping into 12 subsets of SNPs according to the *P* value for data set 1 as follows: P1 ≤ (0.01, 0.05, 0.1, 0.2, 0.3, 0.4, 0.5, 0.6, 0.7, 0.8, 0.9, 1.0). SECA subsequently performs Fisher's exact tests to assess the presence of excess SNPs in which the direction of effects is concordant across data set 1 and data set 2 (that is, for the corresponding *P* value derived 12 subsets of SNPs associated in data set 2, P2). Hence, a total of 144 SNP subsets (a 12 by 12 matrix from data set 1 and data set 2) were assessed for SNP effect concordance. SECA calculates permuted *P* value for the number of significant associations with adjustment for testing 144 associations (based on permutations of 1000 replicates).

In the present study, we first assessed AD GWAS as data set 1 and each of the GIT disorders as data set 2. For comparison, we also assessed each of the GIT disorders as data set 1 against AD as data set 2. Thus, using SECA, we assessed the effects of AD-associated SNPs on each of the GIT disorders and vice versa. Since SECA is conditioned on data set 1, the bi-directional analysis is an important analysis step to account for instances where SNPs that are strongly associated with AD do not affect GIT traits and vice versa. Further, the bi-directional analysis (which is not possible with LDSC, for example) enables the assessment of whether the observed genetic overlap is driven primarily by only one of the traits or both thereby enhancing a better understanding of their association.

**GWAS cross-traits meta-analysis.** GWAS meta-analysis pools the results of GWAS data, thereby increasing the sample sizes and augmenting the detection of genetic variants with small to moderate effect sizes. In the present study, we used the GWAS meta-analysis method of pooling AD GWAS with each of the GIT traits (cross-disorder or cross-trait meta-analysis). We used two models of meta-analysis: the Fixed Effect (FE), and the modified Random Effect (RE2)[82] models. The FE model estimates the FE P-value using the inverse-variance weighted method, which assumes that the AD and each of the GIT disorders' GWAS are assessing the same (fixed) effect. The presence of effect heterogeneity is a limitation of the model. On the other hand, by estimating P-values using the modified random effects, the RE2 model[82] allows for differences in SNP effects and the method is powerful in the presence of SNP effect heterogeneity.

**Genomic loci characterisation.** Using the outputs of our cross-trait meta-analyses for AD and each of the GIT disorders, we carried out some downstream analyses including functional annotation of SNPs, and genomic loci characterisation in line with practice in the previous studies[13,55,83,84]. Briefly, SNPs that were not genome-wide significant in the individual AD and GIT disorder GWAS, but which reached genome-wide significance following the meta-analysis were identified. From these, we characterised independent SNPs at r² < 0.6, and lead SNPs at r² < 0.1. We defined the genomic locus as the region within 250 kb of each lead SNP. We assigned lead SNPs within this region to the same locus, meaning two or more lead SNPs may be present in one locus. We performed these downstream analyses using the Functional Mapping and Annotation (FUMA) software (an online platform)[55]. We subsequently queried identified loci in the GWAS catalogue (https://www.ebi.ac.uk/gwas) and Open Targets Genetics (https://genetics.opentargets.org) to assess their previous identification for AD, GIT disorders or other traits.

**Pairwise GWAS analysis**. We performed a co-localisation analysis utilising the pairwise GWAS (GWAS-PW) method[27] to further assess the regions in the genome shared by AD and GIT disorders. Briefly, GWAS-PW software implements the Bayesian pleiotropy association test and identifies genomic regions that influence a pair of correlated traits[27]. We used this method to assess whether the loci reaching genome-wide significance in our GWAS meta-analyses were truly shared by AD and the GIT disorders. Also, we investigated other shared genomic regions which may not have been found in the GWAS meta-analysis. We combined the summary data for AD with the data for each of the GIT disorders and estimated the posterior probability of association (PPA) of a genomic region using the GWAS-PW software. We modelled four PPAs: (i) that a genomic region is associated with AD only (PPA-1), (ii) that a genomic region is associated with the GIT trait only (PPA-2), (iii) that a genomic region is associated with both AD and the GIT trait and the causal variant is the same (PPA-3), and (iv) that a specific genomic region is associated with both AD and the GIT trait but through separate causal variants (PPA-4)[27].

**Causal relationship assessment**. Using MR[28] analysis methods, we assessed the causal association between AD and each of the GIT disorders in this study. Mimicking randomised control trials (RCTs), MR analysis incorporates genetics into epidemiological study designs to assess causality[28]. The method is based on the principle of instrumental variables and underpinned by three primary assumptions. First is the relevance assumption which requires that the chosen instruments are robustly associated with the exposure variable[85]. Second is the independence assumption which states that the instruments must not be associated with confounders of the exposure-outcome variables[85]. Last is the assumption of exclusion which demands that the instruments influence the outcome only through their relationship with the exposure variable[85].

In the present study, we used the two-sample MR method (https://mrcieu.github.io/TwoSampleMR/articles/introduction.html) for a bidirectional association assessment between AD and each of the GIT disorders. In the first round of analysis (AD as exposure variable), independent ($r^2 < 0.001$) genome-wide significant SNPs ($P < 5 \times 10^{-8}$) associated with AD were utilised as instrumental variables (IVs) and assessed against each of the GIT disorders' GWAS (outcome variables) analysed in this study. This analysis assesses whether genetic predisposition to AD is causally associated with any of the GIT traits included in the present study. Reversing the direction of analysis, independent SNPs robustly associated with each of the GIT disorders' GWAS (exposure variables) were similarly utilised as IVs and assessed against AD (as the outcome variable). In this instance, we assessed the potential causal effects of GIT traits on AD.

We used the inverse variance weighted (IVW) model of MR as the primary method for causal association assessment, and for validity testing, we performed a heterogeneity test (Cochran's Q-test), a 'leave-one-out' analysis, a horizontal pleiotropy check (MR-Egger intercept) and individual SNP MR analyses. Also, we used other MR analysis models including the MR-Egger, weighted median[86,87], and the 'Mendelian randomisation pleiotropy residual sum and outlier' (MR-PRESSO)[88] methods for sensitivity testing. The MR-Egger and weighted median models operate under weaker assumptions of MR and are designed to provide valid causal estimates even when horizontal pleiotropy is present in all (MR-Egger) or as much as 50% (weighted median) of selected IVs[86,87]. Conversely, the MR-PRESSO method can detect and correct horizontal pleiotropy by excluding outlier IVs thereby improving valid causal estimates[88]. All MR analyses were performed in R (4.0.2).

We performed an additional assessment of the causal or partial causal association between AD and each of the GIT disorders using the Latent Causal Variable (LCV) method[29]. LCV estimates causality proportion (GCP) ranging from −1 to 1 where a value close to 1 indicates a potential causal association between two traits in the forward direction and −1 in the backward direction[29]. LCV corrects for heritability and genetic correlation between traits and is not limited by sample overlap[29]. This analysis was performed in the online platform of the Genetics of Complex Traits (CTG) virtual laboratory (https://vl.genoma.io/analyses/lcv)[29,89].

**Gene-based association analysis**. We performed gene-based association analyses to identify genome-wide significant genes shared by both AD and each of the GIT disorders assessed in this study. This analysis complements the SNP-based studies. However, beyond the SNP level, gene-based association analysis provides greater power for identifying genetic risk variants since it aggregates the effects of multiple SNPs, and it is generally not limited by small effect sizes or correlations among SNPs. Moreover, genes are more closely related to biology than SNPs, meaning gene-level analysis can provide better insights into the underlying biological mechanisms of complex traits.

In the present study, we carried out gene-based association analysis separately for AD and GERD using the multi-marker analysis of genomic annotation (MAGMA) software, implemented in the FUMA (https://fuma.ctglab.nl/)[55] platform. We defined gene boundaries length within ±0 kb outside the gene, and to ensure that equivalent gene-based tests were performed, we utilised SNPs overlapping AD and GERD GWAS in analysis separately for each of the traits. Following a similar procedure, we also performed gene-based analysis using SNPs overlapping AD and PGM GWAS.

Based on the results of the gene-based analysis, we identified genome-wide significant genes for each of the traits—AD, GERD and PGM—at an adjusted $P$ value of $2.64 \times 10^{-6}$ (0.05/18929: Bonferroni adjustment for testing 18,929 genes). Further, to identify genes shared by AD and each of GERD and PGM, we extracted their overlapping genes at gene $P$ value <0.1 ($P_{gene} < 0.1$). We combined the respective $P$ values for AD and the GIT traits using Fisher's Combined P-value (FCP) method and thereafter identified shared genes reaching genome-wide significance for AD and each of GERD and PGM in the FCP analyses.

**Pathway-based functional enrichment analysis**. For a better understanding of the potential biological mechanisms underlying AD and GIT disorders or their comorbidity, we carried out pathway-based functional enrichment analyses using the online platform of the g:GOst tool in the g-profiler software[56]. The g:GOst tool performs analysis on the list of user-inputted genes and queries relevant databases including Gene Ontology, Human Protein Atlas, WikiPathway, Human Phenotype Ontology, CORUM, Kyoto Encyclopedia of Genes (KEGG), and Reactome. This analysis enables us to functionally interpret genes overlapping AD and GIT disorders. We included genes that were overlapping between AD and each of GERD and PGM at $P_{gene} < 0.05$ (FCP < 0.02) in this analysis, and followed established protocols[90]. Functional category term sizes were restricted to values from 5 to 350[90]. For multiple testing corrections, we applied the default 'g: SCS algorithm' recommended in the protocol[90] and reported the significantly enriched biological pathways at the multiple testing adjusted $P$ value [$P_{adjusted}$] < 0.05.

**Statistics and reproducibility**. We performed statistical analysis mainly in the Unix environment and the R (https://www.r-project.org/) software. Additional software including Python (https://www.python.org/), Plink (https://www.cog-genomics.org/plink/) and online platforms (CTG virtual lab: https://vl.genoma.io/updates, G-profiler: https://biit.cs.ut.ee/gprofiler, and FUMA: https://fuma.ctglab.nl) were utilised. Adjustment for multiple testing was carried out using the Bonferroni approach in LDSC, gene-based and meta-analyses. In G-profiler, we applied the recommended inbuilt 'g: SCS algorithm' for multiple testing corrections. To enable us to test the reproducibility of AD and GIT association, we used available GIT data for further analysis.

**Ethics approval and consent to participate**. This study is a secondary analysis of existing GWAS summary data from public repositories, and international research consortia. Specific and relevant ethics approval for each of the data utilised is presented in the associated publications described in the section for GWAS summary data. No additional ethics approval is required for the conduct of the present study.

**Reporting summary**. Further information on research design is available in the Nature Research Reporting Summary linked to this article.

# Data availability

All data generated during this study are included in the published article and its Supplementary section. GWAS summary statistics data analysed were sourced from international research consortia and public repositories as described in the subsection for GWAS summary data. The data are freely available and accessible online through the links and references provided within this study. Supplementary Data 1 provides a comprehensive description of the data and how to access them.

# Code availability

We used publicly available software for analysis in this study. Here, we list the URLs (some of which are online methods) for the software where details about them including (where applicable) the computer codes are available: CTG virtual lab (https://vl.genoma.io/updates), FUMA (https://fuma.ctglab.nl/), G-profiler (https://biit.cs.ut.ee/gprofiler/), GWAS Catalogue (https://www.ebi.ac.uk/gwas/home), GWAS-PW (https://github.com/joepickrell/gwas-pw), LDSC (https://github.com/bulik/ldsc), SECA (https://sites.google.com/site/qutsgel/software?authuser=0), Open Target Genetics (https://genetics.opentargets.org/), and Two-Sample MR (https://mrcieu.github.io/TwoSampleMR/articles/introduction.html).

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

## Acknowledgements

The GWAS data analysed in the present study were sourced from several databases and international research groups/consortia. We express our appreciation to all these research groups, consortia, and the UKB for providing us access to their data. This research was supported through funding to S.M.L. from the National Health and Medical Research Council (Australia; APP1161706, APP1191535). Funders have no influence on the design, analysis, or interpretation of findings in this study.

## Author contributions

Conceived the study: E.O.A., T.P., S.M.L.; designed the study: E.O.A., E.K.O., T.P., S.M.L.; conducted the analysis: E.O.A.; interpreted the results: E.O.A., E.K.O., D.R.N., T.P., S.M.L.; drafted the manuscript: E.O.A.; made critical revisions to the manuscript: E.O.A., E.K.O., D.R.N., T.P., S.M.L.; funding: S.M.L. All authors read and approved the final manuscript.

## Competing interests

The authors declare no competing interests.
