## [Peer Review File · Communications Biology]

Reviewers' comments:

Reviewer #1 (Remarks to the Author):

Alzheimer's disease (AD) is a highly polygenic disorder which has been linked to various phenotypes through genetics. In this paper, Adewuyi et al. explores AD's shared genetics with gastrointestinal tract (GIT) disorders. Despite clinical observations of their comorbidity, their link through genetics have so far not been identified. As such this manuscript is relevant and important. Through a variety of statistical genetics approaches, the authors strongly suggest shared genetic architecture.

The main strength of this paper lies in its multiple approaches, from single nucleotide polymorphism (SNP) level analysis, to gene-based analysis to pathway-based analysis. The statistical methods being used are solid. However, as often is the case for multi-level approaches, it is hard to have a highly cohesive take-home message for readers without broad knowledge to combine all levels of analyses, outside of a general relationship between AD and GIT traits. Still, this manuscript may interest specialists at different levels as leads for further investigation.

My biggest concern is on the fact that all GIT traits show shared genetics with AD, excluding IBD (which is likely due to underpowered dataset). While these findings are likely true, the abundance of positive results become less convincing without control datasets. I recommend to repeat the procedures of the SECA and LDSC analyses between GIT traits and (well-powered) control phenotypes, to convince readers of this specificity of AD and GIT. Several candidate phenotypes can be found for instance in Bulik-Sullivan et al., Nature Genetics 2015.

Other notes:

1. (line 161/table 2) The authors claim there is significance concordance of SNP risk effects between AD and all GIT GWAS using SECA method. This is shown by a pattern of increasing strength of association between AD and GERD signified by odds ratio. However, at $P \leq 1$ levels, P Fisher is already at 4.65×10^{-11} (significant). Wouldn't they be at chance levels? This leads one to wonder if these are SNPs are common to any trait and non-specific to GIT.
2. (line 176) It is not clear from this paper alone why one needs to do both direction of analysis between AD and GIT traits as dataset1 and dataset2. Assuming they use the same p-value levels, would it not be the same population of SNPs? I think additional clarity to the methods section on the SECA procedure would help readers without having to look up references.
3. Table 2 is harder to read and includes multiple IBD studies, one not referenced on Table 1. It is best to keep only one dataset which is the higher power, unless required.
4. (Line 159 / line 212) In the SECA procedure, it is not stated that MHC and APOE (a major focal point for AD) were excluded. This procedure was used for LDSC. Is there a reason why SECA does not require exclusion of MHC and APOE?
5. The GIT phenotypes in e.g. Table 3, Figure 2 and Figure 3 and throughout the manuscript are presented in different orders. For example, The first phenotype analysis for SECA/LDSC/meta-analysis is GERD on table 3. Figure 2 shows Peptic ulcer first. Meanwhile the first phenotype analysis for GWAS meta-analysis is PGM. Changing the orders adds additional effort for the reader to compare the results of each GIT disorder according to different methods. I would suggest keeping the same running order of phenotypes for the entire manuscript.
6. Looking into the datasets in Table 1, it appears that IBD study is comparatively massively underpowered, and coincidentally, IBD is the only trait in this study that does not bear significant correlation. It is clearer if Table 1 included effective sample sizes to show this point. A significant amount of space (e.g. line 232-245, line 476-488) is dedicated to explain the lack of IBD association when the space can be dedicated to unpack the positive results.
7. Table 1 sample size is incorrect. IBD 7045 + 426803 is not 456327

8. Table 1 is sorted according to phenotype source, but it is more readable to be sorted by phenotype. Several phenotypes have multiple sources, and it is unclear at various points of the paper which datasets are being used (or whether meta-analyses had been used). For instance, Table 3 refers to two datasets of IBD, one of which is not listed in Table 1.

Reviewer #2 (Remarks to the Author):

Adewuyi et al. presented a comprehensive assessment of the genetic relationship between AD and GIT disorders through analyzing large GWAS summary data at SNP-level, gene-level, and pathway-level. A significant genetic overlap was found between AD and several GIT disorders, several SNPs, genes, pathways are also identified to be shared between AD and GERD.

Overall, the paper is solid and well written. However, I have some concerns about how the statistical genetic analyses approaches were chosen and underlying assumptions. I'd like to see some brief explanation of the ideas underlying these choices in the main text.

Page 7: Why SECA was chosen and the interpretation of reverse analysis results (effect of dataset 1 on dataset 2 and vice versa) will be helpful to the readers.

Page 13: What type of meta-analysis was done and the underlying assumptions? For example, in discussion (page 26), several methods were mentioned "m-value, binary effect, GWAS-PW", how were these approaches chosen, and what are the differences between them?

Page 25: The reason IBD doesn't have significant genetic correlation with other GIT traits could also be related to the relative smaller sample size. This observation might not suggest the different underlying mechanisms between IBD and other GIT disorders.

Page 30: Authors mentioned possible violation of MR underlying assumptions, and several MR models are used, I wonder what are the differences in assumptions between the several approaches used?

Reviewer #3 (Remarks to the Author):

Adewuyi and colleagues investigated the relationship between genetic factors associated with Alzheimer's disease and those associated with gastrointestinal tract (GIT) traits, using data from published, well-powered GWAS.

The authors found (i) a significant genetic overlap and correlation between AD and several types of GIT, (ii) 7 putatively novel loci that reached genome-wide significant in a meta-analysis of AD and GIT, and (iii) several interesting pathways putatively associated with both Alzheimer's diseases and GIT diseases.

The manuscript covers the hot topic of gut-brain axis and its consequences on diseases. Overall, the results of the study are interesting, although not all of them are necessarily novel. The manuscript is well written, and all statistical analyses are well documented in the Methods. However, I am left with the sensation that I'm also not sure what is the main finding of the manuscript. Perhaps the authors should make it more clear.

I have several concerns regarding the SNPs analyses, their discussion and interpretation, that should be addressed prior to publication.

Major comments:

page 8, line 172-181, Results: the authors perform a SECA analysis using several p-value thresholds ($p < 1$, $p < 0.9$, ...) to define the subsets of SNPs to compare between AD and GIT GWAS. It is well-known that in GWAS, the burden of multiple test correction is huge, and I would think that associations with a $p > 5e-5$ (suggestive level of association typically used) should be treated

very carefully as they likely include many false positive associations. Did the author consider to only compare genome-wide significant associations ($p < 5e-8$) or the suggestive associations ($p < 5e-5$)? I feel that this would be a much more reliable comparison than considering all other SNPs.

Page 8, Table 2, Results: the authors report that "a pattern of increasing strength of association between AD and GERD was observed as p-values for the SNPs subsets decrease". However, the comparison of SNPs with the lowest p-value ($p < 0.01$ for both AD and GERD), despite it includes SNPs with more robust association with both AD and GERD, has the highest p-value of all comparisons ($p = 1.47e-2$), and likely does not survive multiple test correction. The authors should discuss this results further and the possible reasons for this.

Page 13, line 268-271, Results: the authors state that none of the 42 SNPs that reached genome-wide significance in the meta-analysis was genome-wide significant in the studies alone (despite being well-powered studies). I think the author should report the association of the "novel" SNPs in the single-studies. I also wonder what is the association (in the meta-analysis) of the SNPs that were genome-wide significant in the single studies alone. The authors should comment on this, especially since they claim that there is a high genetic correlation between AD and GIT traits.

Page 15, Table 4, Results: the authors report that they found 42 novel genome-wide significant SNPs after the meta-analysis. However, many of these novel SNPs rely in well-known regions associated with AD (CR1, HLA, INPP5D, ABI3, PTK2B, CLU and APOE genes), and some are even in LD with the leading SNP associated with AD (e.g. rs530324 ("novel") with the known rs9331896, or rs36133610 ("novel") with the known rs35349669). The authors should discuss the overlap and the LD with previously known SNPs from each of the single studies.

The authors do exclude APOE region during the genetic correlation analysis, yet they include it in the GWAS meta-analysis. It is not clear why this is done, given the well known association of APOE region with AD, and the large LD patterns. Is there a known association of APOE region with GIT traits? After the meta-analysis, the authors found a genome-wide significant association of several SNPs in APOE region, yet the relationship between these and the causal variants for AD are not described much (few lines in discussion) and still treated as novel. I feel that authors should discuss this more carefully.

Minor comments:

page 7, line 169, Results: next to the r^2 value used for LD-pruning, authors should report also the window (in kb) that was used, for reproducibility. This is also not reported in the Methods.

Page 11, line 228, Results: it's not clear if and how the authors controlled for the overlapping samples when doing genetic correlation analyses. They claim that there was no sample overlap, but this looks strange given the similar numbers of controls of the used GWAS, many of which were performed in the UKBiobank.

Page 12, Results: in the cross-trait genetic correlation analyses, I feel that authors should correct the p-values for multiple comparisons.

Page 17, line 311, Results: the authors perform a GWAS meta-analysis between AD and GERD. Given the extremely high genetic correlation between PGM and GERD (0.99), it is not clear why the author performed both meta-analysis and not just one.

Page 21, line 402, Results: the authors used Fisher's method to combine p-values of the gene-based analysis. Did the authors consider to combine p-value weighting the different studies, for example based on the total sample size?

Page 22, Results: the authors perform a functional enrichment analysis using the overlapping (significant) genes (from the gene-based test) between AD and GERD. It's not clear the extent to which the significant pathways that were found, overlap with the pathways that are significant within each single study. For example, lipid metabolism is well known to be involved in AD, and likely also with some GIT traits.

Page 38, Methods: the authors used MAGMA for the gene-based test. However, it is not reported the model adopted and whether they performed any filtering before the test.

Responses to reviewers' comments on manuscript COMMSBIO-21-3489

We appreciate the time and effort that reviewers dedicated to providing insightful comments and constructive feedback on our manuscript. We have thoroughly considered all the comments and revised our manuscript accordingly. We incorporated changes that reflect the detailed suggestions provided. Please, find below our point-by-point response to the comments.

Reviewer #1 (Remarks to the Author):

Comment: The main strength of this paper lies in its multiple approaches, from single nucleotide polymorphism (SNP) level analysis to gene-based analysis to pathway-based analysis. The statistical methods being used are solid.

Response: We thank the reviewer for their constructive feedback and for finding merits in our study.

Comment: However, as often is the case for multi-level approaches, it is hard to have a highly cohesive take-home message.

Response: Yes, we agree and have now provided a clearer and more cohesive conclusion. For example, the conclusion of our abstract now reads as follows:

“Our findings provide genetic insights into the gut-brain relationship, implicating shared but non-causal genetic susceptibility of GIT disorders with AD’s risk. Loci, genes, and biological pathways identified are potential targets for further investigation in AD, GIT disorders, and their comorbidity.”

We also made a similar correction to the conclusion of the manuscript as follows (Lines 582 – 593 of the revised manuscript):

“In conclusion, this study provides genetic insights into the long-standing debate and the observed relationship of AD with GIT disorders, implicating shared genetic susceptibility. Our findings support a significant risk increasing (but non-causal) genetic association between AD and GIT traits (GERD, PUD, PGM, gastritis-duodenitis, IBS, and diverticular disease). Also, we identified genomic regions and genes, shared by AD and GIT disorders that may potentially be targeted for further investigation, particularly, the PDE4B gene (or its subtypes) which has shown promise in inflammatory diseases¹⁻⁴. Our study also underscores the importance of lipid

homeostasis and the potential relevance of statins and lipase inhibitors in AD, GIT disorders or their comorbidity. To our knowledge, this is the first comprehensive study to assess these relationships using statistical genetics approaches. Overall, these findings advance our understanding of the genetic architecture of AD, GIT disorders, and their observed co-occurring relationship.”

Importantly, we provided comprehensive information on the relevance or implications of findings which are important take-home messages (lines 546 – 564 of the revised manuscript).

“Our findings have implications for practice and further studies. First, results highlighting lipid-related mechanisms support the roles of abnormal lipid profiles in the aetiologies of the disorders, which may be potential biomarkers for AD and GIT disorders (or their comorbidity). Second, our findings underscore the importance of lipid homeostasis. The dietary approach is one effective preventive as well as non-pharmacologic approach for the management of hyperlipidaemia, and overall, this is consistent with findings in this study. Indeed, adherence to a ‘Mediterranean’ diet (low in lipids) is recognised as beneficial both in AD⁵ and GIT disorders⁶. Thus, a recommendation for healthy diets, early in life, may form part of the lifestyle modifications for preventing AD and GIT disorders. The clinical utility of these recommendations will need to be further investigated and validated. Third, our study identifies lipase inhibitors and statin pathways in the mechanisms of AD and GIT disorders, which may be a potential therapeutic avenue to explore in the disorders. Hence, we hypothesise that individuals with comorbid AD and GIT traits may gain benefits from these therapies. There is a need to test this hypothesis using appropriate study designs including randomised control trials. Fourth, our study implicates the PDE4B, and given the evidence in the literature^{2-4,7}, we propose that treatment targeted at its inhibition may be promising in comorbid AD and GIT traits. Lastly, while our findings do not necessarily indicate that AD and GIT disorders will always co-occur, they support their shared biology; thus, early detection of AD may benefit from probing impaired cognition in GIT disorders.”

Comment: My biggest concern is on the fact that all GIT traits show shared genetics with AD, excluding IBD (which is likely due to underpowered dataset). While these findings are likely true, the abundance of positive results become less convincing without control datasets. I recommend to repeat the procedures of the SECA and LDSC analyses between GIT traits and (well-powered) control phenotypes, to convince readers of this specificity of AD and GIT.

Response: Yes, we agree that IBD GWAS is comparatively less powerful which may explain its non-significant correlation with AD. Our SECA results lend credence to this position (as

explained in the revised manuscript). We have now performed several analyses to replicate our findings using well-powered GIT summary data. We note that the replication analysis indeed supports our previous findings. For instance, in the LDSC analysis, all the GIT traits except the IBD were positively and significantly correlated with AD (Supplementary Table 3). SECA results were similarly consistent. Thus, the additional analyses have now addressed the concern raised by the reviewer.

We note further that the focus of our study was to carry out an in-depth assessment of the genetic relationship between AD and GIT traits. Thus, we did not assess analyses between GIT traits and other phenotypes apart from AD as that is completely outside the scope of the present study.

Briefly, as requested, we performed the following LDSC and SECA analyses:

1. LDSC:
 - Assess Jansen et al 2019 ⁸ AD GWAS against each of the GIT traits (both discovery and replication sets)
2. SECA:
 - Assess Jansen et al 2019 ⁸ AD GWAS against each of the GIT traits GWAS (both discovery and replication sets)
 - Assess each of the GIT traits GWAS (both discovery and replication sets) against Jansen et al 2019 ⁸ AD GWAS

Table 1 provides a summary of all the data utilised for LDSC and SECA analysis. Tables 2 – 3 present findings of LDSC and SECA analysis, respectively. Replication testing for our findings is reported in Supplementary Tables 3 (LDSC) and 5 (SECA).

Lines 147 – 150 and 158 – 166 of the revised manuscript describe the results of the LDSC analysis. Lines 177 – 185 and 194 – 203 describe SECA results.

Other notes:

1. (Line 161/table 2) The authors claim there is significance concordance of SNP risk effects between AD and all GIT GWAS using SECA method. This is shown by a pattern of increasing strength of association between AD and GERD signified by odds ratio. However, at $P \leq 1$ levels, P Fisher is already at 4.65×10^{-11} (significant). Wouldn't they be at chance

levels? This leads one to wonder if these SNPs are common to any trait and non-specific to GIT.

Response: We note that the findings in SECA are not due to chance, and yes, the SNPs are specific to AD and GIT traits as described in the SECA method. For example, while a considerable number of AD-associated SNPs are expected to overlap with those of GERD, their p-values will differ. SECA's approach of partitioning independent SNPs into a total of 144 subsets (12 by 12 matrix between dataset 1 and dataset 2) is based on the p-value. Hence, concordance of effect direction will not be by chance.

We have now re-written our SECA results to enhance clarity, highlight what is important for readers (in line with our study objectives) and avoid ambiguity or what can be misunderstood/misinterpreted. Please, see Table 3 in the revised manuscript where we summarised SECA findings in a simple, and easy-to-understand manner. Replication testing for SECA is reported in Supplementary Tables 5, and this has similarly been simplified for ease of comprehension.

2. (line 176) It is not clear from this paper alone why one needs to do both direction of analysis between AD and GIT traits as dataset1 and dataset2. Assuming they use the same p-value levels, would it not be the same population of SNPs? I think additional clarity to the methods section on the SECA procedure would help readers without having to look up references.

Response: Analysis in SECA is conditioned on the first data (dataset 1) which will be different in several respects, including sample size (and hence power), from dataset 2. Importantly, the SNPs population will be different since SNPs that are strongly associated with one trait may not necessarily be for the other trait (i.e., p-value informed LD-clumping based on different GWAS produce different sets of independent SNPs)—the rationale for a bidirectional analysis. This is one uniqueness of the SECA method; since analysis is based on dataset 1, we can assess whether the observed genetic association is similarly driven by both or predominantly by one of the datasets.

We have explained a bit more to ensure that readers can easily follow and understand our SECA results without having to refer to the original publication of the method. Please, see the SECA description in the method section of the revised manuscript (lines 645 – 671). We equally provided additional (brief) information on why a bi-directional analysis is conducted in SECA

(lines 668 – 671). Also, we provide concise information about the SECA method in the results section (lines 168 – 176 of the revised manuscript).

3. Table 2 is harder to read and includes multiple IBD studies, one not referenced on Table 1. It is best to keep only one dataset which is the higher power, unless required.

Response: The idea of including two IBD studies was the replication that reviewers emphasised. Since the results for IBD were not significant and so not consistent with findings for other GIT traits, it was necessary to re-assess it using another GWAS. That was the reason we had two studies for IBD in the first submission.

Now, given the emphasis on replication at LDSC and SECA analysis levels, we have reported only one IBD in the main result and use the second one for replication testing.

Importantly, we have re-worked our results for SECA both in the main manuscript (Table 3) and in the Supplementary section (Supplementary Tables 4, 5, and 6), so they are easy to understand.

4. (Line 159 / line 212) In the SECA procedure, it is not stated that MHC and APOE (a major focal point for AD) were excluded. This procedure was used for LDSC. Is there a reason why SECA does not require exclusion of MHC and APOE?

Response: We have now performed SECA with and without the *APOE* and MHC regions (see Table 3, Supplementary Table 5, etc).

We note that the exclusion of these regions did not qualitatively change our results.

5. The GIT phenotypes in e.g. Table 3, Figure 2 and Figure 3 and throughout the manuscript are presented in different orders. For example, the first phenotype analysis for SECA/LDSC/meta-analysis is GERD on table 3. Figure 2 shows Peptic ulcer first. Meanwhile the first phenotype analysis for GWAS meta-analysis is PGM. Changing the orders adds additional effort for the reader to compare the results of each GIT disorder according to different methods. I would suggest keeping the same running order of phenotypes for the entire manuscript.

Response: We thank the reviewer for this observation. We have now maintained a consistent order in the Tables and other aspects of the manuscript.

6. Looking into the datasets in Table 1, it appears that IBD study is comparatively massively underpowered, and coincidentally, IBD is the only trait in this study that does not bear significant correlation. It is clearer if Table 1 included effective sample sizes to show this point.

Response: We agree with the reviewer that the IBD GWAS is comparatively less powerful. We have calculated the effective sample sizes for each of the data, as recommended. Given Table 1 is already overloaded, we summarised details of the effective sample size calculations in Supplementary Table 1, where datasets utilised in this study are more comprehensively described.

A significant amount of space (e.g. line 232-245, line 476-488) is dedicated to explain the lack of IBD association when the space can be dedicated to unpack the positive results.

Response: We have further summarised the discussion as requested. See lines 484 – 491 (now 8 lines in this revised manuscript as opposed to the 12 in the initial submission).

7. Table 1 sample size is incorrect. IBD 7045 + 426803 is not 456327

Response: We thank the reviewer for bringing this mistake to our attention. We have made corrections as necessary. The controls sample size is 449,282 and not 426,803. This has now been corrected.

IBD: $7045 + 449,282 = 456327$

8. Table 1 is sorted according to phenotype source, but it is more readable to be sorted by phenotype. Several phenotypes have multiple sources, and it is unclear at various points of the paper which datasets are being used (or whether meta-analyses had been used). For instance, Table 3 refers to two datasets of IBD, one of which is not listed in Table 1.

Response: We have re-ordered Table 1 and it is now sorted by phenotype, as requested, to make it more readable. Also, we have ensured that datasets used for analysis are easy to identify.

Issues around two IBD datasets have similarly been resolved as we used the second IBD data for replication testing. Please, see Table 1 in the revised manuscript. Thanks

Reviewer #2 (Remarks to the Author):

Comment: Adewuyi et al. presented a comprehensive assessment of the genetic relationship between AD and GIT disorders through analyzing large GWAS summary data at SNP-level, gene-level, and pathway-level. A significant genetic overlap was found between AD and several GIT disorders, several SNPs, genes, pathways are also identified to be shared between AD and GERD. Overall, the paper is solid and well written.

Response: We thank the reviewer for their constructive feedback and for finding merits in our manuscript.

However, I have some concerns about how the statistical genetic analyses approaches were chosen and underlying assumptions. I'd like to see some brief explanation of the ideas underlying these choices in the main text.

Response: We have now provided explanations as requested. For example, we provided information summarising how and why our analysis was conducted in lines 108 – 121 of the revised manuscript, with brief information on the ideas supporting the choice of the methods. For example, we stated our objectives and chose methods that will enable us to achieve them (lines 108 – 121). Also, in the method section, we have justified why we used each of the methods applied in this study (see the methods section generally). Additional information for the SECA method (lines 645 – 671 of the revised manuscript), for example, and even assumptions underlying MR analysis have been summarised (lines 715 – 720 of the revised manuscript). We provide more details in response to comments about specific analytical methods below.

Page 7: Why SECA was chosen and the interpretation of reverse analysis results (effect of dataset 1 on dataset 2 and vice versa) will be helpful to the readers.

Response: We appreciate the merit of this comment and have provided additional explanations as requested. Please, see the SECA description in the method section of the revised manuscript

(lines 645 – 671). We provide additional (brief) information on why a bi-directional analysis is conducted in SECA (lines 666 – 671).

“Since SECA is conditioned on dataset 1, the bi-directional assessment is an important analysis step to account for instances where SNPs that are strongly associated with AD do not affect GIT traits and vice versa. Further, the bi-directional analysis (which is not possible with LDSC, for example) enables the assessment of whether the observed genetic overlap is driven primarily by only one of the traits or both thereby enhancing a better understanding of their association.”

Briefly, we chose SECA because the method provides an additional approach to assessing and understanding the SNP-level genetic relationship between AD and GIT disorders (the focus of our study). So, in addition to using the LDSC method in assessing genetic correlation, we used SECA which enables us to understand concordance in the direction of effects and whether one trait increases the risk for the other or contrariwise. Importantly, unlike LDSC, SECA enables the assessment of whether the observed association is driven primarily by only one of the traits or both, that is, by performing bi-directional analyses.

Interpretation of SECA results (and reverse analysis) is more about dataset 1 upon which analysis is conditioned. Where there is a significant concordance of SNP risk effect, for instance, we would interpret this as SNPs strongly associated with dataset 1 equally influence the trait represented by dataset 2. Where both directions are significant, we would be confident to say there is evidence of strong genetic overlap between the two. Since analysis is conditioned on dataset 1, it is possible to have an instance where only one direction is significant in which case we’ll need to know why, for example, it may be due to underpowered data as in the case of IBD in the present study. We thank the reviewer for this comment and trust that it has now been well addressed.

Page 13: What type of meta-analysis was done and the underlying assumptions? For example, in discussion (page 26), several methods were mentioned “m-value, binary effect, GWAS-PW”, how were these approaches chosen, and what are the differences between them?

Response: We aim to identify shared genomic loci and we used several approaches to enable reliable results—consistent findings across the methods mean increased confidence in the results. First, we used a GWAS meta-analysis approach and implemented fixed effect (FE) and

modified RE (RE2)⁹ models. The RE2 handles heterogeneity (a limitation in the FE model) and is powerful to detect association⁹. We have now removed the ‘m-value’ and binary effect to avoid the possibility of our work being misunderstood as we believe they do not add any additional value. Lastly, we used colocalisation analysis (GWAS-PW¹⁰ method) to assess whether loci identified in meta-analyses are truly shared by AD and GIT traits and to identify additional loci not picked in the meta-analysis. We have described all these approaches in the methods section of our manuscript (Meta-analysis—lines 673 – 683, GWAS-PW—lines 697 – 710 of the revised manuscript).

Page 25: The reason IBD doesn’t have significant genetic correlation with other GIT traits could also be related to the relative smaller sample size. This observation might not suggest the different underlying mechanisms between IBD and other GIT disorders.

Response: We agree that the correlation results for IBD may largely be because of the comparatively smaller sample size of the data, and we have endeavoured to concisely emphasise this fact in our discussion section. Please, see the first paragraph of our discussion section (lines 484 – 492 of the revised manuscript).

“... In contrast to the positive genetic correlation between AD and other GIT disorders, LDSC found no significant genetic correlation between AD and IBD, which may be due to the relatively small number of cases and sample size of the IBD GWAS. Supporting this premise, SECA revealed a significant association between AD (as dataset 1) against IBD (as dataset 2), but not the other way around. The AD GWAS has a larger sample size, providing a more robust association on which to condition (select independent) SNPs for concordance analysis which may explain why the significant association was not bi-directional. Future studies, nonetheless, need to confirm this relationship, as more powerful IBD GWAS becomes available.”

Page 30: Authors mentioned possible violation of MR underlying assumptions, and several MR models are used, I wonder what are the differences in assumptions between the several approaches used?

Response: We have provided more information about the assumptions underlying the two-sample MR (lines 715 – 720 of the revised manuscript).

“... The method is based on the principle of instrumental variables and underpinned by three primary assumptions. First is the relevance assumption which requires that the chosen instruments are robustly associated with the exposure variable¹¹. Second is the independence

assumption which states that the instruments must not be associated with confounders of the exposure-outcome variables¹¹. Last is the assumption of exclusion which demands that the instruments influence the outcome only through their relationship with the exposure variable¹¹.”

Also, for the various models used in our study, we have concisely described their respective differences or underlying principles (lines 736 – 741 of the revised manuscript).

“The MR-Egger and weighted median models operate under weaker assumptions of MR and are designed to provide valid causal estimates even when horizontal pleiotropy is present in all (MR-Egger) or as much as 50% (weighted median) of selected IVs^{12,13}. Conversely, the MR-PRESSO method can detect and correct horizontal pleiotropy by excluding outlier IVs thereby improving valid causal estimates¹⁴.”

Reviewer #3 (Remarks to the Author):

The manuscript covers the hot topic of gut-brain axis and its consequences on diseases. Overall, the results of the study are interesting, although not all of them are necessarily novel. The manuscript is well written, and all statistical analyses are well documented in the Methods.

Response: We thank the reviewer for their constructive feedback and for finding merits in our manuscript.

However, I am left with the sensation that I’m also not sure what is the main finding of the manuscript. Perhaps the authors should make it more clear.

Response: We have re-written the conclusion of this study to enhance ease of understanding main findings and a clearer “take-home” message, as equally suggested by reviewer 1. To avoid duplicating the lengthy response provided, please, see our response to the comment of the first reviewer (first part of this document). Thanks

I have several concerns regarding the SNPs analyses, their discussion and interpretation, that should be addressed prior to publication.

Major comments:

page 8, line 172-181, Results: the authors perform a SECA analysis using several p-value thresholds ($p < 1$, $p < 0.9$, ...) to define the subsets of SNPs to compare between AD and GIT GWAS. It is well-known that in GWAS, the burden of multiple test correction is huge, and I would think that associations with a $p > 5e-5$ (suggestive level of association typically used) should be treated very carefully as they likely include many false positive associations. Did the author consider to only compare genome-wide significant associations ($p < 5e-8$) or the suggestive associations ($p < 5e-5$)? I feel that this would be a much more reliable comparison than considering all other SNPs.

Response: We thank the reviewer for their comment and appreciate the importance of multiple testing corrections in our type of analysis. We note, however, that biases of multiple testing do not occur here. SECA is based on testing a matrix of 12 by 12 independent SNP subsets from a pair of GWAS summary data, resulting in 144 associations. The major focus was not suggestive or genome-wide significant SNPs but all the overlapping independent SNPs (to have a total of 144 associations).

Moreover, SECA uses Fisher's test to calculate permuted p-value (permutation of 1000 replicates conducted) for the number of significant associations with adjustment for testing a total of 144 SNP subsets. Our results are based on the permuted p-values. So, multiple testing correction is inherently captured in the association results. We have updated the SECA method description to reflect this understanding (lines 645 – 671 of the revised manuscript).

For example, in lines 660 – 66, we stated:

"... SECA calculates permuted P-value for the number of significant associations with adjustment for testing 144 association (based on permutations of 1000 replicates)."

Similarly, in lines 180 – 182, we stated:

"... The empirical P-values ($P_{permuted}$) for the significant associations, adjusting for the 144 SNP subsets tested (using permutations of 1000 replicates)."

Page 8, Table 2, Results: the authors report that "a pattern of increasing strength of association between AD and GERD was observed as p-values for the SNPs subsets decrease". However, the comparison of SNPs with the lowest p-value ($p < 0.01$ for both AD and GERD), despite it includes SNPs with more robust association with both AD and GERD, has the highest p-value

of all comparisons ($p=1.47e-2$), and likely does not survive multiple test correction. The authors should discuss these results further and the possible reasons for this.

Response: As explained in our response to an earlier comment, there is no bias of multiple testing in our SECA results (see our response to the comment above). We understand that the way our SECA results were initially presented did come across quite clearly. To prevent a likely misunderstanding of these results, we have now presented our SECA findings in an easy to comprehend manner (see Table 3, and Supplementary Tables 3 – 5).

Page 13, line 268-271, Results: the authors state that none of the 42 SNPs that reached genome-wide significance in the meta-analysis was genome-wide significant in the studies alone (despite being well-powered studies). I think the author should report the association of the “novel” SNPs in the single-studies. I also wonder what is the association (in the meta-analysis) of the SNPs that were genome-wide significant in the single studies alone. The authors should comment on this, especially since they claim that there is a high genetic correlation between AD and GIT traits.

Response: First, the reviewer requested that we ‘report the association of the “novel” SNPs in the single studies’. This we already did in Table 4 (see columns with the following headings: ‘meta-analysis’, ‘AD’ and ‘GIT disorders’). Second, the reviewer requested ‘the association (in the meta-analysis) of the SNPs that were genome-wide significant in the single studies alone’. We have provided this information (where there is an association following a meta-analysis) in Supplementary Tables 10).

For the 42 SNPs reaching genome-wide significance for AD and PGM meta-analysis, we stated that the SNPs were not genome-wide significant in the individual AD or PGM GWAS, that is before the meta-analysis ($5 \times 10^{-8} < P_{\text{GWAS-data}} < 0.05$). Given the merit of this comment, we have re-assessed the identified SNPs thoroughly and more carefully. While they were not genome-wide significant in the individual studies, some of them are in loci previously reported for AD or GIT disorders. We have made needed corrections in our results and the discussion section. For example, for AD vs GERD meta-analysis, please, see lines 217 – 233 of the revised manuscript.

“First, a meta-analysis of AD and GERD identified a total of 119 SNPs reaching genome-wide significant association ($P_{\text{meta-analysis}} < 5 \times 10^{-8}$, Supplementary Table 8), from which we

characterised seven independent ($r^2 < 0.1$) genomic loci—1p31.3, 1q31.1, 3p21.31, 6p21.32, 17q21.32, 17q21.33, 19q13.32 (Table 4). Many SNPs reaching genome-wide significance in these loci were not genome-wide significant in the individual AD and GIT GWAS we analysed but reached the status in the cross-trait (pairwise) meta-analyses (Table 3). The observation that some of these loci are known for AD or GIT traits provides support for our cross-trait analysis findings. Specifically, two of the identified loci: (1p31.3 [near PDE4B], and 3p21.31 [near SEMA3F]) were not previously genome-wide significant for AD, indicating they are putatively novel for the disorder. Similarly, three of the seven loci: (17q21.32 [ZNF652], 17q21.33 [PHB], and 19q13.32 [TOMM40, APOC2, KLC3, ERCC2]) are putatively novel for GERD given they were not previously genome-wide significant for the disorder. A locus at 1q31.1 (near BRINP3) was putatively novel for both AD and GERD at the time of our analysis but has now been reported in a recent GERD multi-trait analysis¹⁵—providing support for our finding. The remaining locus, 6p21.32 (near genes HLA-DQA2 and HLA-DRA) is known for both AD¹⁶ and GIT disorders—IBD¹⁷, ulcerative colitis¹⁸, and Crohn's disease¹⁷, and now (in our study) for GERD.”

Also, for AD and PGM, please see lines 261 – 277 of the revised manuscript.

“... we performed a meta-analysis of PGM with AD thereby identifying 42 SNPs (Supplementary Table 14) at seven independent loci (Table 4) reaching a genome-wide significance level. This analysis replicated, at a genome-wide level ($P_{\text{meta-analysis}} < 5 \times 10^{-8}$), five of the seven genome-wide loci found in the AD and GERD meta-analysis including 1p31.3, 3p21.31, 6p21.32, 17q21.33 and 19q13.32. Additional loci found in the AD and PGM meta-analyses such as 16q22.1 and 1q32.2 were at least genome-wide suggestive ($P_{\text{meta-analysis}} < 1 \times 10^{-5}$) in the AD and GERD analysis, supporting their involvement in the disorders. An additional 23 SNPs, at three loci, were genome-wide suggestive ($P_{\text{meta-analysis}} < 1 \times 10^{-5}$) in the AD and PGM meta-analysis (Supplementary Table 15). Of these, the rs33998678 SNP (16q22.1, IL34) is in strong LD ($r^2 = 0.91$) with a genome-wide significant locus (rs34644948, at 16q22.1, MTSS2, Table 4) found in the AD vs PGM analysis, providing more support for its involvement in both traits. Similarly, the rs663576 SNP (at 17q21.32, PHOSPHO1) is moderately correlated ($r^2 = 0.41$) with a genome-wide significant SNP (rs2584662 at 17q21.33, PHB, Table 4), identified in the meta-analysis. This locus (17q21.33) was found in AD and GERD meta-analysis (SNP rs2584662 near PHB), lending support for its involvement in AD and the GIT traits. Supplementary Table 10 summarises the sentinel AD loci associated with PGM and vice versa.”

In addition, many loci reached only genome-wide suggestive association levels and we have reported those appropriately (Supplementary Tables 9, 13, and 15). These were also described in the text (lines 234 – 242, 259 – 261, and 269 – 278 of the revised manuscript).

Page 15, Table 4, Results: the authors report that they found 42 novel genome-wide significant SNPs after the meta-analysis. However, many of these novel SNPs rely in well-known regions associated with AD (CR1, HLA, INPP5D, ABI3, PTK2B, CLU and APOE genes), and some are even in LD with the leading SNP associated with AD (e.g. rs530324 (“novel”) with the known rs9331896, or rs36133610 (“novel”) with the known rs35349669). The authors should discuss the overlap and the LD with previously known SNPs from each of the single studies.

Response: We appreciate the meticulous assessment of the reviewer and have carefully and more closely gone through our analysis and results with the view of making corrections where necessary. As noted in our response to the comment above, some of the identified loci are known for AD while some are known for GIT traits. We have updated our manuscript appropriately, for example, see lines 217 – 233 of the revised manuscript for the AD vs GERD meta-analysis results).

The authors do exclude APOE region during the genetic correlation analysis, yet they include it in the GWAS meta-analysis. It is not clear why this is done, given the well known association of APOE region with AD, and the large LD patterns.

Response: We excluded the *APOE* region for genetic correlation analysis to be sure that any observed association (correlation) was not driven primarily by the region that is well-established for AD. Interestingly with or without the region, we found a positive and significant genetic correlation between AD and GIT traits. The pattern of results was replicated using the SECA method.

In the case of GWAS meta-analysis, the inclusion of the *APOE* region cannot bias our results in any way, since the assessment was about individual SNPs and we can see which SNPs are improved or not improved by the analysis. Also, the inclusion of the region does not in any way influence other regions or the significance or otherwise of our findings. Indeed, our meta-analysis and colocalisation findings indicate that the region is shared by both AD and GIT traits, hence, it is appropriate and important not to exclude it in the GWAS meta-analysis.

...Is there a known association of APOE region with GIT traits? After the meta-analysis, the authors found a genome-wide significant association of several SNPs in APOE region, yet the relationship between these and the causal variants for AD are not described much (few lines in discussion) ... and still treated as novel. I feel that authors should discuss this more carefully.

Response: We did not treat the *APOE* as being novel for AD and GIT disorder in this study. We noted that it has been well-established for AD (*results section: lines 274 – 276 of the initial manuscript*).

And, yes, there is evidence associating *APOE* genotype with the gut microbiome¹⁹, and IBD^{20,21}. We mentioned this in the discussion section of the initial manuscript submitted:

“The remaining locus (19q13.32), harbouring the APOE gene, has a well-established association with AD, and our results suggest it is also involved in these GIT phenotypes. A previous association of APOE genotype with the gut microbiome¹⁹, and IBD^{20,21} may support current findings” (lines 511 – 514 of the initial manuscript).

We have now provided more information about the identified loci and described traits previously reported to be associated with them (lines 279 – 313 of the revised manuscript). For example, in lines 300 – 304, we stated as follows:

“Further, our analysis consistently identified and replicated the 19q13.32 locus (mapped genes: TOMM40, APOC2, KLC3, ERCC2, BCL3, and CD33) as shared by AD and GIT disorders. While this locus is well known for AD, notably, it has similarly been linked with GIT traits including IBD²² (SYMPK, lead SNP: rs16980051, GRCh37: 19:46,345,886), and gut microbiota²³, thus, highlighting an association of AD with not only GIT disorders, but also the gut microbiome.”

We believe that our extensive revision fully addresses the comments of the reviewer.

Minor comments:

page 7, line 169, Results: next to the r^2 value used for LD-pruning, authors should report also the window (in kb) that was used, for reproducibility. This is also not reported in the methods.

Response: We have included more detailed information, including the kb for clumping, in the description of the SECA approach (lines 652 – 655 in the methods section of the revised manuscript). For example, concerning the window of clumping, we stated thus:

“...Second we performed two rounds of P-value informed LD clumping in dataset 1 (first clumping: -clump-r² 0.1, -clump-kb 1000; second clumping: -clump-r² 0.1, -clump-kb 10000) using PLINK 1.90²⁴.”(lines 708 – 711 of the revised manuscript).

Page 11, line 228, Results: it’s not clear if and how the authors controlled for the overlapping samples when doing genetic correlation analyses. They claim that there was no sample overlap, but this looks strange given the similar numbers of controls of the used GWAS, many of which were performed in the UKBiobank.

Response: We used LDSC in assessing the genetic covariance intercept²⁵. The results of this analysis (Supplementary Table 2) indicates that the genetic covariance intercept for the analysis of AD with each of the GIT trait was not significantly different from zero, indicating no significant sample overlap.

Page 12, Results: in the cross-trait genetic correlation analyses, I feel that authors should correct the p-values for multiple comparisons.

Response: We have used Bonferroni adjustment for multiple testing corrections.

“We applied Bonferroni adjustment for testing the effects of seven GIT traits on AD ($0.05/7 = 7.1 \times 10^{-3}$), and all genetic correlation results surviving this cut-off were considered significant while those having $P < 0.05$ were regarded as nominally significant.” (Lines 641 – 644 of the revised manuscript).

Page 17, line 311, Results: the authors perform a GWAS meta-analysis between AD and GERD. Given the extremely high genetic correlation between PGM and GERD (0.99), it is not clear why the author performed both meta-analyses and not just one.

Response: Performing GWAS meta-analysis between AD and each of GERD and PGM enabled a potential replication of our results.

Page 21, line 402, Results: the authors used Fisher’s method to combine p-values of the gene-based analysis. Did the authors consider to combine p-value weighting the different studies, for example based on the total sample size?

Response: No, we did not use weighting in our analysis. However, given the interesting nature of this comment/suggestion, we have now used Stouffer's (equal weights/equal sample sizes) as well as the weighted Stouffer's methods to compare and clarify this concern for AD-GERD

analysis, for example. We note that the results of these methods were the same as those of Fisher's combined p-value method. Please, see Supplementary Table 24 for this clarification.

Page 22, Results: the authors perform a functional enrichment analysis using the overlapping (significant) genes (from the gene-based test) between AD and GERD. It's not clear the extent to which the significant pathways that were found, overlap with the pathways that are significant within each single study. For example, lipid metabolism is well known to be involved in AD, and likely also with some GIT traits.

Response: We acknowledge that pathway analysis has the potential to be over-interpreted. Hence, in this study, we were more interested in characterising genes overlapping AD and GIT traits using pathway-based analysis and not necessarily in identifying pathways overlapping from the individual GWAS. Thus, overlapping genes (as described in the method section) were utilised for pathway-based analysis, thereby, identifying the significantly enriched biological pathways found in our study. We followed a well-established protocol for this analysis²⁶ and ensured that equivalent gene-based tests were performed for each of the pair of traits by utilising only SNPs overlapping in their respective GWAS. While this was not part of the objective of the present study, given the reviewer's comment, we have now performed the analysis and found not many of the pathways for the individual GERD and AD GWAS overlapped (Supplementary Tables 27 and 28).

Page 38, Methods: the authors used MAGMA for the gene-based test. However, it is not reported the model adopted and whether they performed any filtering before the test.

Response: We used MAGMA as implemented in the FUMA platform with the default settings as described in our method section (lines 805 – 819 of the revised manuscript).

References

- 1 Li, H., Zuo, J. & Tang, W. Phosphodiesterase-4 Inhibitors for the Treatment of Inflammatory Diseases. *Front Pharmacol* **9**, 1048-1048, doi:10.3389/fphar.2018.01048 (2018).
- 2 Spadaccini, M., D'Alessio, S., Peyrin-Biroulet, L. & Danese, S. PDE4 Inhibition and Inflammatory Bowel Disease: A Novel Therapeutic Avenue. *Int J Mol Sci* **18**, 1276, doi:10.3390/ijms18061276 (2017).
- 3 Richter, W., Menniti, F. S., Zhang, H.-T. & Conti, M. PDE4 as a target for cognition enhancement. *Expert Opin Ther Targets* **17**, 1011-1027, doi:10.1517/14728222.2013.818656 (2013).

- 4 Sanders, O. & Rajagopal, L. Phosphodiesterase Inhibitors for Alzheimer's Disease: A Systematic Review of Clinical Trials and Epidemiology with a Mechanistic Rationale. *J Alzheimers Dis Rep* **4**, 185-215, doi:10.3233/ADR-200191 (2020).
- 5 van den Brink, A. C., Brouwer-Brolsma, E. M., Berendsen, A. A. M. & van de Rest, O. The Mediterranean, Dietary Approaches to Stop Hypertension (DASH), and Mediterranean-DASH Intervention for Neurodegenerative Delay (MIND) Diets Are Associated with Less Cognitive Decline and a Lower Risk of Alzheimer's Disease—A Review. *Advances in Nutrition* **10**, 1040-1065, doi:10.1093/advances/nmz054 (2019).
- 6 Elmaliklis, I.-N. *et al.* Increased Functional Foods' Consumption and Mediterranean Diet Adherence May Have a Protective Effect in the Appearance of Gastrointestinal Diseases: A Case-Control Study. *Medicines (Basel)* **6**, 50, doi:10.3390/medicines6020050 (2019).
- 7 Tibbo, A. J. & Baillie, G. S. Phosphodiesterase 4B: Master Regulator of Brain Signaling. *Cells* **9**, 1254 (2020).
- 8 Jansen, I. E. *et al.* Genome-wide meta-analysis identifies new loci and functional pathways influencing Alzheimer's disease risk. *Nature genetics* **51**, 404-413 (2019).
- 9 Han, B. & Eskin, E. Random-effects model aimed at discovering associations in meta-analysis of genome-wide association studies. *The American Journal of Human Genetics* **88**, 586-598 (2011).
- 10 Pickrell, J. K. *et al.* Detection and interpretation of shared genetic influences on 42 human traits. *Nature genetics* **48**, 709 (2016).
- 11 Davies, N. M., Holmes, M. V. & Smith, G. D. Reading Mendelian randomisation studies: a guide, glossary, and checklist for clinicians. *BMJ* **362**, k601 (2018).
- 12 Bowden, J., Davey Smith, G., Haycock, P. C. & Burgess, S. Consistent estimation in Mendelian randomization with some invalid instruments using a weighted median estimator. *Genetic epidemiology* **40**, 304-314 (2016).
- 13 Bowden, J., Davey Smith, G. & Burgess, S. Mendelian randomization with invalid instruments: effect estimation and bias detection through Egger regression. *International journal of epidemiology* **44**, 512-525 (2015).
- 14 Verbanck, M., Chen, C.-y., Neale, B. & Do, R. Detection of widespread horizontal pleiotropy in causal relationships inferred from Mendelian randomization between complex traits and diseases. *Nature genetics* **50**, 693-698 (2018).
- 15 Ong, J.-S. *et al.* Multitrait genetic association analysis identifies 50 new risk loci for gastro-oesophageal reflux, seven new loci for Barrett's oesophagus and provides insights into clinical heterogeneity in reflux diagnosis. *Gut*, doi:10.1136/gutjnl-2020-323906 (2021).
- 16 Schwartzenuber, J. *et al.* Genome-wide meta-analysis, fine-mapping and integrative prioritization implicate new Alzheimer's disease risk genes. *Nat Genet* **53**, 392-402, doi:10.1038/s41588-020-00776-w (2021).
- 17 de Lange, K. M. *et al.* Genome-wide association study implicates immune activation of multiple integrin genes in inflammatory bowel disease. *Nat Genet* **49**, 256-261, doi:10.1038/ng.3760 (2017).
- 18 Anderson, C. A. *et al.* Meta-analysis identifies 29 additional ulcerative colitis risk loci, increasing the number of confirmed associations to 47. *Nat Genet* **43**, 246-252, doi:10.1038/ng.764 (2011).
- 19 Parikh, I. J. *et al.* Murine Gut Microbiome Association With APOE Alleles. *Frontiers in Immunology* **11**, doi:10.3389/fimmu.2020.00200 (2020).
- 20 Al-Meghaiseeb, E. S. *et al.* Genetic association of apolipoprotein E polymorphisms with inflammatory bowel disease. *World journal of gastroenterology: WJG* **21**, 897 (2015).
- 21 Glapa-Nowak, A. *et al.* Apolipoprotein E variants correlate with the clinical presentation of paediatric inflammatory bowel disease: A cross-sectional study. *World J Gastroenterol* **27**, 1483-1496, doi:10.3748/wjg.v27.i14.1483 (2021).

- 22 Liu, J. Z. *et al.* Association analyses identify 38 susceptibility loci for inflammatory bowel disease and highlight shared genetic risk across populations. *Nature Genetics* **47**, 979-986, doi:10.1038/ng.3359 (2015).
- 23 Bonder, M. J. *et al.* The effect of host genetics on the gut microbiome. *Nat Genet* **48**, 1407-1412, doi:10.1038/ng.3663 (2016).
- 24 Purcell, S. *et al.* PLINK: a tool set for whole-genome association and population-based linkage analyses. *The American journal of human genetics* **81**, 559-575 (2007).
- 25 Bulik-Sullivan, B. K. *et al.* LD Score regression distinguishes confounding from polygenicity in genome-wide association studies. *Nature Genetics* **47**, 291-295, doi:10.1038/ng.3211 (2015).
- 26 Reimand, J. *et al.* Pathway enrichment analysis and visualization of omics data using g:Profiler, GSEA, Cytoscape and EnrichmentMap. *Nature Protocols* **14**, 482-517, doi:10.1038/s41596-018-0103-9 (2019).

REVIEWERS' COMMENTS:

Reviewer #1 (Remarks to the Author):

The authors have made changes according to this reviewer's suggestions, answering most points of concern and made the paper easier to read. Of important note, changes on the abstract and conclusion have made a clearer take home message of the paper.

Regarding my main point of concern about control phenotypes, my recommendations might have not been well-conveyed nor understood. It is not the power of GIT datasets (in particular IBD) that is the concern, but it is that the paper lacks any control phenotypes outside of AD. The validity of AD and GIT is not being questioned here, however, it is not clear whether the AD-GIT relationship is specific i.e. whether the methods in this paper would reveal similar effects for random pairings of GIT and random phenotypes. The authors chose that this is outside of the scope of the project, which is fair, however if a control phenotype can be added, this would strengthen the result.

1. The authors have added information on the methods.
2. The authors have rewritten the methods to be more understandable to readers
3. The two IBD studies have been handled correctly.
4. The analysis with exclusion of MHC and APOE has been sufficiently performed.
5. The writing order of the phenotypes is now easier for readers.
6. The authors have added effective sample sizes in supplementary table, but it is not very visible. It merits pointing out IBD in Results or Discussion, that IBD is not only "relatively small number of cases" (line 161), but in actuality one magnitude underpowered to other GIT datasets.
7. The table content is now corrected.
8. The table reads much better.

Reviewer #2 (Remarks to the Author):

All my comments have been addressed. Thanks

Reviewer #3 (Remarks to the Author):

The authors provide a revised version of the manuscript extensively addressing my concerns about some statistical issues, transparency, and clarity.

As a result, the manuscript greatly improved in readability and the message of the paper is now more clear.

I have no other issues that should be addressed.

Response to Reviewers' comments

We thank the reviewers for the time and effort dedicated to reviewing our manuscript. We found their comments helpful, and these have contributed to improving the quality of our work. Below we provide a point-by-point response to their comments in this version of our manuscript.

Reviewer #1 (Remarks to the Author):

The authors have made changes according to this reviewer's suggestions, answering most points of concern and made the paper easier to read. Of important note, changes on the abstract and conclusion have made a clearer take home message of the paper.

Response: We thank the reviewer for their constructive feedback. We are happy that our previous edits and changes addressed the reviewer's concerns. Thank you for finding merits in our study.

Regarding my main point of concern about control phenotypes, my recommendations might have not been well-conveyed nor understood. It is not the power of GIT datasets (in particular IBD) that is the concern, but it is that the paper lacks any control phenotypes outside of AD. The validity of AD and GIT is not being questioned here, however, it is not clear whether the AD-GIT relationship is specific i.e. whether the methods in this paper would reveal similar effects for random pairings of GIT and random phenotypes. The authors chose that this is outside of the scope of the project, which is fair, however if a control phenotype can be added, this would strengthen the result.

Response: We thank the reviewer for providing this clarification. We note that our study was undertaken with a specific hypothesis/question about the potential shared genetics of AD and GIT disorders; hence, these are the traits we focused on. We used methods and performed several analyses which enabled us to achieve the stated objectives. Importantly, the analysis approaches utilised have been proven to be robust with well-controlled type-I error rates. Also, some of the approaches are robust to sample overlap and population structure/stratification. To the best of our knowledge, there are no instances in the literature where false positives have been found using these approaches; so, we are confident in our findings. We thank the reviewer for this comment but wish to maintain the analysis as it is.

1. The authors have added information on the methods.

Response: Very many thanks

2. The authors have rewritten the methods to be more understandable to readers

Response: Very many thanks.

3. The two IBD studies have been handled correctly.

Response: Very many thanks.

4. The analysis with exclusion of MHC and APOE has been sufficiently performed.

Response: Very many thanks.

5. The writing order of the phenotypes is now easier for readers.

Response: Very many thanks.

6. The authors have added effective sample sizes in the supplementary table, but it is not very visible. It merits pointing out IBD in Results or Discussion, that IBD is not only a “relatively small number of cases” (line 161) but in actuality one magnitude underpowered to other GIT datasets.

Response: We have added this information in the results and discussion sections:

‘Our estimates of effective sample size (Supplementary Data 1) suggest the IBD GWAS was underpowered compared to other GIT datasets’ (lines 138 – 139 of the revised manuscript).

‘Based on the effective sample size estimates, the IBD GWAS is underpowered compared to other GIT datasets’ (lines 369 – 370 of the revised manuscript).

7. The table content is now corrected.

Response: Thanks

8. The table reads much better.

Response: Thanks

Reviewer #2 (Remarks to the Author):

All my comments have been addressed. Thanks

Response: We thank the reviewer for their constructive feedback. Thank you for finding merits in our study.

Reviewer #3 (Remarks to the Author):

The authors provide a revised version of the manuscript extensively addressing my concerns about some statistical issues, transparency, and clarity.

As a result, the manuscript greatly improved in readability and the message of the paper is now more clear.

I have no other issues that should be addressed.

Response: We thank the reviewer for their constructive feedback. Thank you for finding merits in our study.